Resource

# Genome-wide R-loop analysis defines unique roles for DDX5, XRN2, and PRMT5 in DNA/RNA hybrid resolution

Oscar D Villarreal[1,]*, Sofiane Y Mersaoui[1,2,]*, Zhenbao Yu[1,]*, Jean-Yves Masson[2], Stéphane Richard[1]

**DDX5, XRN2, and PRMT5 have been shown to resolve DNA/RNA hybrids (R-loops) at RNA polymerase II transcription termination sites at few genomic loci. Herein, we perform genome-wide R-loop mapping using classical DNA/RNA immunoprecipitation and high-throughput sequencing (DRIP-seq) of loci regulated by DDX5, XRN2, and PRMT5. We observed hundreds to thousands of R-loop gains and losses at transcribed loci in DDX5-, XRN2-, and PRMT5-deficient U2OS cells. R-loop gains were characteristic of highly transcribed genes located at gene-rich regions, whereas R-loop losses were observed in low-density gene areas. DDX5, XRN2, and PRMT5 shared many R-loop gain loci at transcription termination sites, consistent with their coordinated role in RNA polymerase II transcription termination. DDX5-depleted cells had unique R-loop gain peaks near the transcription start site that did not overlap with those of siXRN2 and siPRMT5 cells, suggesting a role for DDX5 in transcription initiation independent of XRN2 and PRMT5. Moreover, we observed that the accumulated R-loops at certain loci in siDDX5, siXRN2, and siPRMT5 cells near the transcription start site of genes led to antisense intergenic transcription. Our findings define unique and shared roles of DDX5, XRN2, and PRMT5 in DNA/RNA hybrid regulation.**

## Introduction

R-loops are three-stranded structures consisting of a DNA/RNA hybrid and the displaced strand of single-stranded DNA. R-loops are typically, but not exclusively, formed co-transcriptionally where there is reannealing of the nascent transcript to its complementary DNA template with an estimated frequency of 5% depending on the locus and its sequence, transcription levels, and overall gene length (Sanz et al, 2016; Stork et al, 2016; Wahba et al, 2016). These R-loops are mainly associated with accessible chromatin at the transcription start sites (TSS) of gene promoters and at the transcription termination sites (TTS) (Manzo et al, 2018). R-loops formed

at the TSS are predominantly guanine-cytosine (GC)-rich content favoring RNA polymerase pausing and initiation at these CpG promoters (Manzo et al, 2018). R-loops also play an important role in RNA polymerase II processivity with its associated cofactor TFII S during elongation (Zatreanu et al, 2019) and have been shown to function as reversible superhelical stress relievers during transcription (Stolz et al, 2019). Transcription termination is a major event where R-loops are formed. The release of the nascent RNA requires processing by the cleavage and polyadenylation complex and the remaining RNA polymerase II–associated RNA is removed by helicases and digested by the 5′-3′ exonuclease XRN2 (Skourti-Stathaki et al, 2011). R-loops are associated with the establishment of heterochromatin and DNA methylation, especially at CpG promoters (Chedin, 2016; Sanz et al, 2016). The formation of R-loops has been shown to be necessary for the antisense transcription of long noncoding RNAs neighboring an RNA polymerase II gene (Tan-Wong et al, 2019). R-loops are necessary for immunoglobulin class switch recombination targeting activation-induced cytidine deaminase (AID) to the IgH S-regions (Yu et al, 2003; Ribeiro de Almeida et al, 2018).

R-loops pose a major threat for the cell as the lack of their resolution can lead to DNA breaks causing genomic instability (Skourti-Stathaki & Proudfoot, 2014; Aguilera & Gomez-Gonzalez, 2017). R-loops increase the potential for genotoxic transcription–replication collisions, leading to DNA polymerase stalling and collapse of replication forks with the production of DNA breaks (Crossley et al, 2019). In humans, increased R-loops are associated with a variety of diseases that exhibit genomic instability, including myelodysplastic syndromes, neurological disorders, and cancer (Richard & Manley, 2017; Wells et al, 2019).

The processing of the nascent RNAs into mRNAs and their export into the cytoplasm requires the coordinated effort of numerous factor and machineries such as splicing factors, RNA processing, and export machineries (Lukong et al, 2008). If these RNA processing events are defective, unscheduled R-loops accumulate at certain genomic loci (Li & Manley, 2005; Aguilera & Gomez-Gonzalez, 2017).

[1]Segal Cancer Center, Lady Davis Institute for Medical Research and Gerald Bronfman Department of Oncology and Departments of Biochemistry, Human Genetics and Medicine, McGill University, Montréal, Canada   [2]Genome Stability Laboratory, Centre Hospitalier Universitaire de Québec Research Center, Oncology Axis; Department of Molecular Biology, Medical Biochemistry and Pathology; Laval University Cancer Research Center, Québec City, Canada

Correspondence: stephane.richard@mcgill.ca
*Oscar D Villarreal, Sofiane Y Mersaoui, and Zhenbao Yu contributed equally to this work

The clearance and the prevention of unscheduled R-loops are tightly regulated by enzymes, including topoisomerases, helicases, and RNases. DNA topoisomerases such as Top1 and Top3B alter the topology of DNA by removing transcription-induced negative super-coiling to restrict access of the newly transcribed RNA transcripts to its complementary genomic DNA (Tuduri et al, 2009; El Hage et al, 2010; Yang et al, 2014; Manzo et al, 2018). In addition, when formed, R-loops can be removed by cellular RNA nucleases, including RNase H1 and RNase H2, which specifically cleave the RNA moiety of the R-loops (Wahba et al, 2011) and the 5′-3′ exoribonuclease 2 (XRN2) which degrades nascent RNA downstream the 3′-terminal cleavage site to facilitate transcription termination. XRN2 physically associates with helicase such as Senataxin (Skourti-Stathaki et al, 2011) and DDX5 (Mersaoui et al, 2019). The RNA helicases resolve the DNA/RNA hybrids for subsequent degradation by XRN2 (Skourti-Stathaki et al, 2011; Mersaoui et al, 2019). Deficiency of XRN2 expression causes R-loop accumulation and DNA damage (Morales et al, 2016; Mersaoui et al, 2019).

DExD/H RNA helicases are known to be implicated in various aspects of RNA metabolism, including regulation of R-loop accumulation (Tanner & Linder, 2001). Multiple members of the DExD/H helicase family are significantly enriched in cellular DNA/RNA hybrid interactome (Cristini et al, 2018; Wang et al, 2018) and are able to resolve R-loops in vitro using their helicase activity (Hodroj et al, 2017; Song et al, 2017; Chakraborty et al, 2018; Ribeiro de Almeida et al, 2018; Mersaoui et al, 2019). Suppression of cellular R-loops by this family of helicases is functionally associated with the preservation of genome integrity (Hodroj et al, 2017; Song et al, 2017; Cristini et al, 2018), transcription promotion (Argaud et al, 2019), transcription termination (Cristini et al, 2018; Mersaoui et al, 2019) and class switch recombination (Ribeiro de Almeida et al, 2018). In contrast, some members of this family of helicases, such as DDX1 and DHX9, can also promote R-loop formation by resolving RNA secondary structures, including stable guanine quadruplexes to facilitate pairing of the linearized RNA with its complementary DNA (Chakraborty et al, 2018; Ribeiro de Almeida et al, 2018).

A variety of different families of helicases including senataxin (SETX) (Wahba et al, 2011; Chang et al, 2017), RNA helicase aquarius (AQR) (Sollier et al, 2014), Bloom syndrome protein (BLM) (Chang et al, 2017), PIF1 (Tran et al, 2017), and several members of the DExD/H families of RNA helicases (Li et al, 2016; Hodroj et al, 2017; Sridhara et al, 2017; Ribeiro de Almeida et al, 2018; Tedeschi et al, 2018) including DDX5 (Mersaoui et al, 2019) accumulate R-loops in cells when they are deficient.

Little is known about how posttranslational modifications regulate R-loop metabolism. Accumulating evidence indicates that protein arginine methylation plays an important role in R-loop metabolism (Yang et al, 2014; Zhao et al, 2016; Mersaoui et al, 2019). Protein arginine methylation is catalyzed by a family of nine protein arginine methyltransferases (PRMTs) (Bedford & Clarke, 2009). PRMT5 is a type II enzyme that catalyzes the symmetrical arginine dimethylation of protein substrates (Guccione & Richard, 2019). PRMT5 has multiple cellular functions, and its depletion causes aberrant RNA splicing, DNA damage, R-loop accumulation, and genomic instability in multiple cell types (Yang

& Bedford, 2013; Guccione & Richard, 2019). Our previous study demonstrated that PRMT5 methylation of DDX5 at its RGG/RG motif was required for interaction with XRN2 to resolve R-loops at TTS regions (Mersaoui et al, 2019). In the present article, we define the genome-wide loci where R-loops accumulate in the absence of DDX5, XRN2, or PRMT5.

# Results

## DDX5-, XRN2-, and PRMT5-deficient U2OS cells accumulate R-loops genome-wide

Previously, we showed that DDX5-, XRN2-, and PRMT5-deficient cells accumulate R-loops at the TTSs of specific loci in U2OS cells (Mersaoui et al, 2019). Herein, we performed DNA/RNA immunoprecipitation (IP) and high-throughput sequencing (DRIP-seq) to identify the genome-wide R-loops regulated by DDX5, XRN2, and PRMT5. U2OS cells were transfected with siRNAs for DDX5, XRN2, and PRMT5, and the knockdown of each was confirmed by immunoblotting using β-ACTIN as a loading control (Fig 1A). Briefly, we followed the protocol of Chedin and coworkers (Ginno et al, 2012), where genomic DNA was digested with five restriction enzymes to maintain the integrity of the DNA/RNA duplexes. R-loops were subsequently immunoprecipitated using the anti-DNA/RNA hybrid–specific S9.6 antibody (Boguslawski et al, 1986; Phillips et al, 2013), and the DNA strand of the DNA/RNA hybrid was sequenced. A negative control sample, in which the extracted genomic DNA was digested with RNase H before the S9.6 IP, was included to control for DNA/RNA hybrids versus nonspecific DNA duplexes. We observed 50,650 consensus peaks among the siLuciferase (siCTL), siDDX5, siXRN2, and siPRMT5 conditions, covering a total of 135 megabases (Mb) representing ~4.5% of the genome (Supplemental Data 1). Consensus peaks were obtained by taking the union of the peaks identified in each condition relative to its corresponding input. Each pair of biological replicates (denoted by replicates A and B) showed a high correlation in peak intensity (Fig S1A; Pearson Correlation Coefficient of >0.98). Most peaks were RNase H sensitive defining them as DNA/RNA hybrids (Supplemental Data 1 and Fig S1B, top two tracks; Fig S1C and D, right panels). The number of total peak reads in all three knockdown conditions was higher than that in the control (Fig 1B), demonstrating that the loss of either DDX5, XRN2, or PRMT5 causes an accumulation of cellular R-loops. Representative loci by Integrative Genomic Viewer (IGV) are shown, where R-loops at RFNG, GPS1, and DUS1L genes increased in DDX5, XRN2, and PRMT5 knockdown cells (Fig 1C). We observed R-loop peak gains with DDX5 deficiency at previously reported loci, including EGR1, MALAT1, PRMT7, LINC01346, NFKIL2, SLC25A3, JUN, and EEF1A1 (Supplemental Data 1, [Mersaoui et al, 2019]). Interestingly, we also identified specific loci for each knockdown condition, for example, FOS for siDDX5; SSTR5-AS1 and SSTR5 for siPRMT5; and SPIB and MYBPC2 for siXRN2, and the representative IGVs of the gain peaks are shown in Fig S1B. We confirmed by DRIP-PCR the R-loop gains in DDX5 depleted cells at FOS as well as the KLF2, JUNB, CTNNB1, LY6E, SNHG12, SOWAHC, and RPS23 loci (Fig S2).

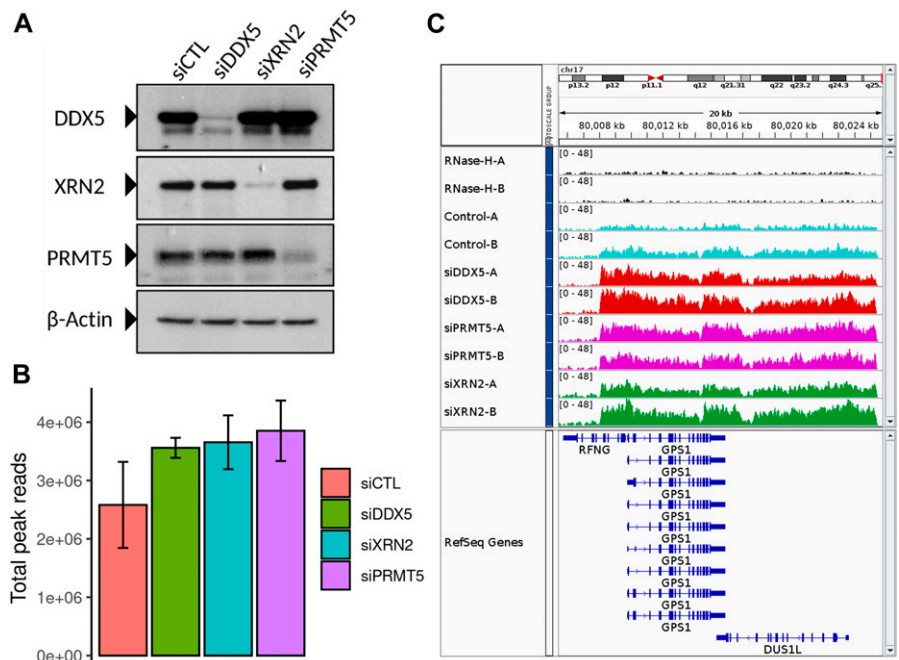

**Figure 1. Genome-wide R-loops in DDX5, XRN2, and PRMT5 knockdown cells.**
**(A)** U2OS cells were transfected with siRNAs for control (CTL), DDX5, XRN2, and PRMT5 and cell lysates were separated by SDS–PAGE and immunoblotted with the indicated antibodies to confirm successful knockdowns. β-Actin was used as a loading control. **(B)** Total read counts within R-loop peaks in each knockdown condition, normalized to library size, and averaged for the two biological replicates. Peaks called by the MACS algorithm v2.2.6 in broad mode (q-value < 0.1) for each replicate were merged into a consensus list across all treatments through DiffBind v2.14.0. Error bars denote SD of the replicates. **(C)** Read coverage of a representative peak profile with a gain in R-loop signal relative to siCTL at the *RFNG-GPS1-DUS1L* gene loci for all cells, generated by Integrative Genomic Viewer v2.8.0. DNA extracted the control DNA treated with RNase-H (black); siCTL cells (cyan); siDDX5 (red); siPRMT5 (magenta); and siXRN2 (green).

Statistical analysis identified 762, 1,059, and 2,632 R-loop peaks with significant increase in intensity (termed gain peaks, q-value < 0.1 and log-fold-change > 1) covering 5.33, 5.70, and 15.29 Mb in DDX5-, XRN2-, and PRMT5-deficient cells, respectively (Figs 2A–C and S1C and D, Table 1, and Supplemental Data 1). In addition, 89, 48, and 145 R-loop peaks had reduced signals (termed loss peaks) spanning 0.43, 0.18, and 0.40 Mb (Figs 2A–C and S1C and D, Table 1, and Supplemental Data 1). DDX5-deficient cells had 184 and 345 gain peaks that overlapped with the gain peaks of XRN2- and PRMT5-deficient cells, respectively (Fig 2B, $P < 6.4 \times 10^{-140}$ and $<1.3 \times 10^{-235}$, Fisher's exact test), suggesting that these three proteins coordinately control cellular R-loop accumulation at specific genomic loci. Moreover, the XRN2-depleted cells had 480 overlapping gain peaks with PRMT5-knocked down cells (Fig 2B, $P < 4.9 \times 10^{-324}$). A significant overlap of loss peaks in these three conditions was identified as well (Fig 2B; $P < 2.6 \times 10^{-23}$, $< 1.1 \times 10^{-13}$, and $<1.8 \times 10^{-29}$ for siDDX5-siXRN2, siDDX5-siPRMT5, and siXRN2-siPRMT5, respectively).

### R-loop accumulation is not a consequence of increased transcription in DDX5-, XRN2-, and PRMT5-deficient cells compared to control

To determine whether the increased R-loop accumulation in the DDX5-, XRN2-, and PRMT5-deficient cells was simply caused by an increase in RNA gene expression, we defined the transcriptomic changes of DDX5-, XRN2-, and PRMT5-depleted U2OS cells compared with siluciferase (siCTL) cells using RNA sequencing (RNA-seq). Unsupervised clustering analysis based on principal component analysis of sample correlations show high reproducibility among the replicates, with the first two PCs explaining 76% of the variance (Fig S3A and B). The expression of 389, 697,

and 1,268 genes were up-regulated versus 406, 970, and 835 genes down-regulated in siDDX5-, siXRN2-, and siPRMT5-transfected U2OS cells, respectively (Fig S3C and Supplemental Data 2).

We matched each peak from the DRIP-seq to an overlapping or the nearest gene and verified the transcriptomic expression as defined by RNA-seq (Supplemental Data 1). Importantly, the RNA expression of overlapping or nearest genes which showed increase in R-loop peak signals in DDX5-, XRN2-, and PRMT5-deficient cells did not significantly vary between control and knockdown cells (Fig S3D; ANOVA $P = 0.76$, 0.91, and 0.31, respectively). This was further quantified using Venn diagrams and, for example, of the siDDX5 697 genes containing at least one R-loop gain peak, only 10 and 15 overlapping or nearest genes were down-regulated and up-regulated for RNA expression in DDX5-deficient cells, respectively (Fig S3E). Indeed, very little overlap was observed in the R-loop gains and losses with gene expression down-regulated and up-regulated across samples (Fig S3E). Altogether, these results show that increased R-loop accumulation does not correlate with increased transcription in DDX5, XRN2, and PRMT5 knockdown cells compared with control cells.

### DDX5, XRN2, and PRMT5 repress R-loop accumulation at chromosome regions with high gene density

The 762 peaks corresponding to siDDX5 R-loop gains were closer to a neighboring gene than the unchanged R-loop bulk peaks ($P < 2.22 \times 10^{-16}$, t test, Fig 3A), implying that they lie in gene-rich areas. Conversely, the siDDX5 resultant R-loop losses (89 peaks) tended to be further from a neighboring gene than the average of the unchanged R-loop peaks ($P = 5.8 \times 10^{-5}$, t test, Fig 3A). XRN2-deficient cells also depicted R-loop gains and losses peaks that were closer

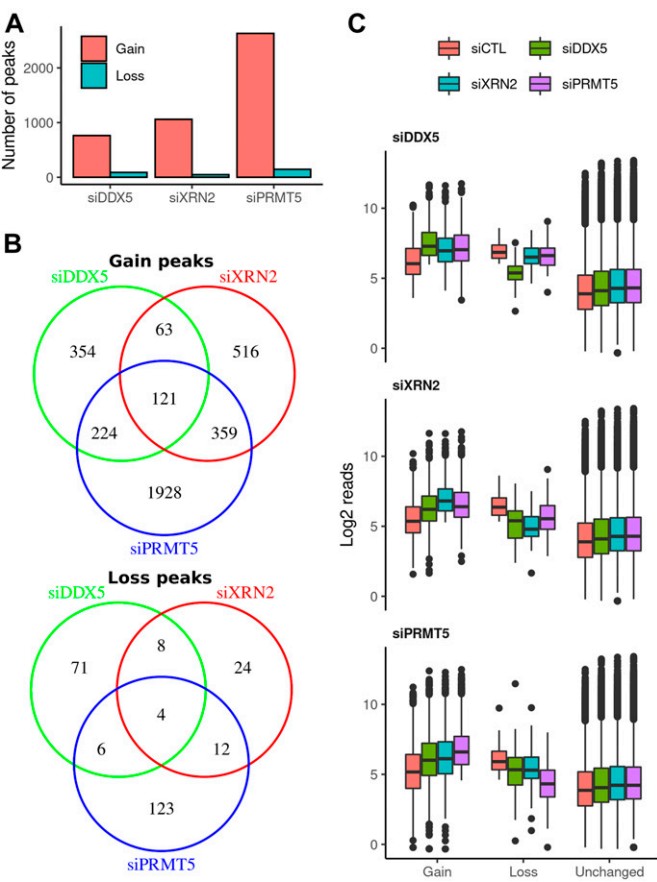

**Figure 2. R-loop gains and losses in siDDX5, siXRN2, and siPRMT5 cells.**
**(A)** Total amount of R-loop gain and loss consensus peaks called for each knockdown condition relative to siCTL by DESeq2 v1.26.0 (absolute log$_2$ fold change > 1 and false discovery rate < 0.1, Wald test). **(B)** Venn diagrams showing the overlaps among consensus peaks with gain (top) or loss (bottom) in R-loop signal upon each knockdown condition. **(C)** Distribution of log$_2$ normalized read concentration within R-loop gain, loss or unchanged consensus peaks at the control (red) and treated (siDDX5, green; siXRN2, cyan; and siPRMT5, purple) cells. Peaks are split into three panels for each knockdown treatment.

and further to a neighboring gene than the unchanged R-loops, respectively ($P < 2.22 \times 10^{-16}$ and $P = 0.027$, $t$ test). The 2,632 R-loop gain peaks of siPRMT5 were closer to a neighboring gene compared

**Table 1. R-loop genomic coverage.**

| siRNA | Type | Space (Mb) |
|---|---|---|
| DDX5 | Gain | 5.33 |
| DDX5 | Loss | 0.43 |
| DDX5 | Unchanged | 129.26 |
| XRN2 | Gain | 5.70 |
| XRN2 | Loss | 0.18 |
| XRN2 | Unchanged | 129.14 |
| PRMT5 | Gain | 15.29 |
| PRMT5 | Loss | 0.40 |
| PRMT5 | Unchanged | 119.33 |

to the unchanged R-loop bulk peaks ($P < 2.22 \times 10^{-16}$, $t$ test, Fig 3A). However, the PRMT5 R-loop loss peaks showed similar distances to neighboring genes than unchanged R-loops ($P < 0.16$, $t$ test, Fig 3A). The R-loop gains in cells deficient for DDX5, XRN2, and PRMT5 were more prevalent in chromosomes with very high gene density, for example, chromosomes 1, 19, and 20, whereas R-loop gains minimally occurred on chromosomes with a very low gene density, that is, chromosomes 4 and 18 (Fig 3B). DDX5 R-loop losses were unaffected by the density of genes on chromosomes and did not show a noticeable trend (Fig 3B). In contrast, XRN2 and PRMT5 R-loop losses occurred more frequently on chromosomes with a low gene density, that is, chromosomes 4 and 18 (Fig 3B). In conclusion, DDX5, XRN2, and PRMT5 play a more important role in R-loop repression at genomic regions with higher transcription intensity.

### DDX5-, XRN2-, and PRMT5-deficient cells increase R-loops at TTSs

Analysis of the 46,839 DRIP-seq peaks (a subset of the 50,650 total peaks) identified without a gain or loss in signal relative to siCTL in any of the samples revealed that more than half of the peaks (62.9%) were located in intronic regions, representing nascent transcripts, whereas 22.6% were located near defined noncoding and coding genes at the promoter-TSS, exons including 5′-UTR and 3′-UTR, and the TTS (Fig 4A; Unchanged pie chart). Interestingly, 6.4% (3% 3′UTR and 3.4% TTS) of the peaks were at 3′ end of the annotated genes in the 46,839 unchanged DRIP-seq peaks identified, and this increased to 9.5% (3.3% 3′UTR; 6.2% TTS) for siDDX5, 12.0% (5.2% 3′UTR; 6.8% TTS) for siXRN2, and 10.6% (4.6% 3′UTR; 6.0% TTS) for siPRMT5 gain peaks (Fig 4A). Further analysis revealed that the gain peaks overlapping in siDDX5 and siXRN2 (6.0% 3′UTR; 12.0% TTS), siDDX5 and siPRMT5 (4.3% 3′UTR; 8.4% TTS), siXRN2 and siPRMT5 (5.8% 3′UTR; 8.1% TTS), and all three (6.6% 3′UTR; 11.6% TTS) were more enriched at the 3′-terminal region (Fig 4B), suggesting a coordinate regulation of R-loops at the 3′-terminus of genes by DDX5, XRN2, and PRMT5. Examination of the distribution of the peak reads showed a higher number from the TSS to the TTS in cells deficient for DDX5, XRN2, or PRMT5 than siCTL cells (Fig 4C). Subtraction of siCTL reads shows a better distribution of the peak reads of DDX5, XRN2, and PRMT5 samples (Fig 4D). In the case of DDX5 and XRN2, the peaks were mainly concentrated at the TSS and TTS with less peaks in the gene bodies (Fig 4D). Interestingly, for siPRMT5 cells, there was a gradual increase in peak reads from the TSS to the TTS (Fig 4D). We next removed all the reads within the gene bodies revealing an increase in intergenic reads upstream and downstream of the TSS and TTS for DDX5, XRN2, and PRMT5 compared with siCTL (Fig 4E), and this distribution was similar for DDX5, XRN2, and PRMT5 when subtracting siCTL reads (Fig 4F). Last, we examined only the intronic R-loop peaks for siDDX5, siXRN2, and siPRMT5 cells. The intronic reads for DDX5-, XRN2-, or PRMT5-depleted cells were highly clustered before the TTS, whereas intronic reads near the TSS were mainly present in siDDX5 and siXRN2 cells (Fig 4G and H). The subtraction of siCTL revealed a striking pattern of intronic reads for siPRMT5 cells being low near the TSS and accumulating to high levels at the TTS (Fig 4H). Overall, most R-loop peak reads

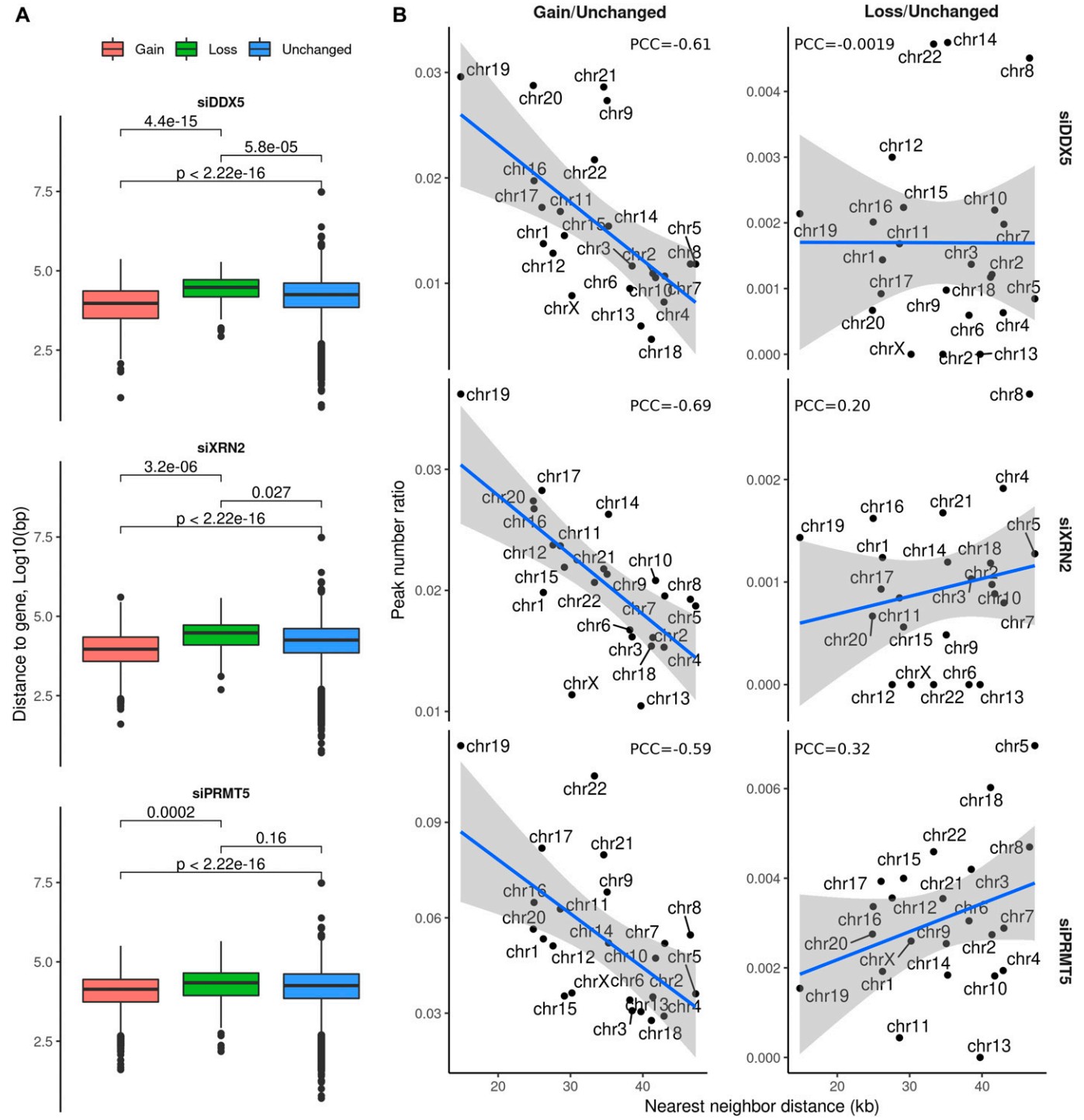

**Figure 3.   R-loop gains are elevated close to neighboring genes on chromosomes with high gene density.**
**(A)** Distribution of distance to the neighboring gene from the DRIP-seq consensus peaks with a gain (red), loss (green), or unchanged (blue) R-loop signal upon each siRNA condition, measured through Bedtools v2.26.0 with Ensembl gene annotation. Definition of neighboring gene is the second nearest gene to the peak. **(B)** Ratio between the total amount of R-loop gain or loss peaks and the amount of unchanged peaks upon knockdown treatment, as a function of the mean distance from all consensus peaks to their corresponding neighboring (i.e., second nearest) genes, measured separately for each chromosome.

were distributed equally in siDDX5, siXRN, and siPRMT5 cells, especially at the TTS, consistent with a coordinated role in transcription termination. The unique read peak patterns in siPRMT5 cells, especially the gradual increase in intronic peak reads from the TSS to TTS, suggests PRMT5 has DDX5- and XRN2-independent roles in nascent RNA regulation.

Figure 4. Distribution and coverage of DRIP-seq peaks across genomic locations.
(A) Distribution of the peaks with no change in R-loop signal upon any of the knockdown treatments (left; Unchanged) and of the gain peaks upon each of the siDDX5, siXRN2, and siPRMT5 cells. The percentage of the R-loop gain peaks are distributed as 5′-UTR and 3′-UTR; promoter and transcription start site (promoter-TSS); transcription termination site (TTS), and noncoding, intronic, and intergenic peaks. (B) Distribution in percentage of the gain peaks in the intersection of two of three of the knockdown treatments (left) or in all of the knockdown treatments simultaneously (right). (C, D, E, F, G, H) Genome-wide R-loop signal profile for each treatment

## DDX5-, XRN2-, and PRMT5-deficient cells regulate R-loop formation to different levels at transcription initiation sites

Although the percentage of R-loop gain peaks at the 3′-terminus (3′-UTR + TTS) increased in DDX5-, XRN2-, and PRMT5-deficient cells, the R-loop gain peaks at the 5′-terminus (promoter-TSS and 5′-UTR) relative to the unchanged peaks (8.2% + 1.0%, Fig 4A) showed an increase in siDDX5-deficient (13.5% + 1.2%) and XRN2-deficient cells (9.9% + 0.9%) and was lower in PRMT5-deficient cells (5.7% + 0.5%, Fig 4A). In fact, PRMT5-deficient cells had 1,067 absolute R-loop gains at the TSS, whereas DDX5-deficient and XRN2-deficient cells had 428 and 420, respectively (Fig 4I). Thus, it is the elevated peak reads at the TTS for PRMT5-deficient cells (Fig 4D) that gives the apparent lower percentage in Fig 4A at the TSS. We then divided the R-loop peaks into two groups in which the center of an R-loop is closer to nearest TSS or TTS. In comparison with the unchanged peaks (i.e., those without a gain in any condition), the siDDX5 and siXRN2 gain peaks both lied significantly closer to the nearest TSS ($P = 2.8 \times 10^{-11}$ and $P = 1.1 \times 10^{-11}$, $t$ test) or to the TTS ($P = 1.5 \times 10^{-11}$ and $P < 2.22 \times 10^{-16}$, $t$ test, Fig S4A and B). PRMT5-deficient gain peaks lied significantly closer to the nearest TTS (Fig S4B, $P < 2.22 \times 10^{-16}$), but further from TSS (Fig S4A, $P < 1.7 \times 10^{-06}$). Importantly, the siDDX5 gain peaks were more likely to be located closer to the TSS (56%) than to the TTS (44%), whereas the siXRN2 and siPRMT5 gains behaved in an opposite manner (40% and 41% near TSS and 60% and 59% near TTS, respectively, Fig 4I). Of the 428 DDX5 R-loop gain peaks near the TSS, we distributed them into 10 groups (Fig 4J); all peaks at the TSS and TTS; R-loop gain peaks from siDDX5 cells alone near the TSS or TTS; R-loop gain peaks from siDDX5 that overlap with R-loop gain peaks from siXRN2 cells near the TSS or TTS; R-loop gain peaks from siDDX5 that overlap with R-loop gain peaks from siPRMT5 cells near the TSS or TTS; R-loop gain peaks from siDDX5 that overlap with R-loop gain peaks from siXRN2; and R-loop gain peaks from siPRMT5 cells near the TSS or TTS. The group with the highest percentage (~75%) was siDDX5 cells alone near the TSS, suggesting a role for DDX5 at the TSS independent of XRN2 and PRMT5. The other group with the highest percentage was siDDX5, siXRN2, and siPRMT5 at the TTS (Fig 4J), further indicating a coordinated role in transcription termination.

Inspection of the nucleotide sequence of the R-loops generated by siDDX5, siXRN2, or siPRMT5 gain peaks show an increased GC-rich content compared with all peaks (Fig S4C and D). These observations are consistent with the siDDX5 R-loop gain peaks enriched at TSS and being rich in CG-rich areas (CpG islands; Fig S4E), known to be symmetrically distributed near the TSS of promoters (Saxonov et al, 2006). These observations implicate a role for the R-loops generated by DDX5-, XRN2-, and PRMT5-deficiency in transcription regulation of CpG island genes.

## Increased antisense intergenic transcription neighboring R-loop gain peaks located at the TSS

Recent studies reported that R-loops have the capability to act as inherent promoters for RNA polymerase II transcription initiation and facilitate the synthesis of antisense long noncoding RNA occurring frequently at gene promoter regions upstream of the TSS (Tan-Wong et al, 2019). We searched for RNA-seq peaks potentially representing antisense transcripts neighboring the TSS upstream of 2 kb (Supplemental Data 1; columns BT–CB) and TTS downstream of 2 kb (Supplemental Data 1; columns CC–CK). Genomic loci were identified with an increase in intergenic RNA-seq reads upstream of the TSS or downstream of the TTS of genes near to the R-loop gain peaks in siDDX5, siXRN2, and siPRMT5 cells using the threshold described in the Materials and Methods section (Supplemental Data 3). The unique list of loci for each siDDX5, siXRN2, and siPRMT5 cells (Supplemental Data 3) suggest that the proteins function independently of each other for this function. For DDX5, we selected eight loci for further reverse transcription (RT)-PCR confirmation of the increase of intergenic RNA. IGV tracks showed that *EGR1*, *ACTG1*, *RHOB*, and *UBALD1* had increases of RNA-seq intergenic reads upstream of the TSS and downstream of the TTS, whereas *RB1CC1*, *SOGA1*, and *STIL* had increases of RNA-seq intergenic reads only upstream of the TSS in DDX5-deficient cells (Fig 5A). Finally *IER2* only had increase of RNA-seq intergenic reads downstream of the TTS (Fig 5A). We next performed RT-qPCR comparing RNA levels between siCTL and siDDX5 cells using random hexamers for RT to normalize the RNA levels to GAPDH mRNA. In siDDX5-depleted U2OS cells, we confirmed a statistical increase in the intergenic RNA levels neighboring the eight genes selected (Fig 5B), consistent with the RNA-seq data (Supplemental Data 2 and Fig 5A). To confirm whether the intergenic RNA reads were derived from antisense transcription, we used semi-quantitative RT-PCR with sense, antisense, or random primers for the RT reaction. We amplified a DNA fragment corresponding to the antisense intergenic RNA upstream of the TSS neighboring the *EGR1*, *ACTG1*, *RHOB*, *RB1CC1*, *SOGA1*, *STIL*, and *UBALD1* genes using sense primers for the RT reaction (Fig 5C). As a control, a DNA fragment was amplified for the intergenic region downstream of the TTS for *IER2* with an antisense primer, demonstrating this transcript is of sense orientation and likely represents transcription read-through passed the TTS (Fig 5C). For siXRN2 and siPRMT5, we selected four loci for IGV visualization. IGV tracks for siXRN2 loci included *RPLP1*, *RPP25L*, *FBXW5*, and *PMEPA1* (Fig S5). IGV tracks for PRMT5 loci included *CUL3*, *NYAP1*, *CPT1A*, and *NACC2* (Fig S5). All these loci showed an increase in intergenic RNA transcripts neighboring the TSS, representing potential antisense transcripts. Taken together, these findings show that increased accumulation of R-loops near the TSS of certain loci in DDX5-, XRN2-, and PRMT5-deficient cells induces intergenic antisense RNA expression.

# Discussion

In the present article, using classical DRIP-seq, we define the genome-wide analysis of R-loop alterations in cells deficient of DDX5, XRN2, and PRMT5. More than 50,650 DRIP-seq peaks were

---

compared with control (C, E, G) and normalized to siCTL (D, F, H). **(C, D, E, F, G, H)** DRIP-seq coverage was measured using all reads (C, D), reads overlapping with intergenic regions (E, F), and reads overlapping with introns (G, H) through NGS.PLOT v2.63. **(I)** Histogram of peaks lying nearer to the TSS or to the TTS of the nearest gene using the full list of consensus peaks (left) or the gain peaks relative to the control (right) measured through HOMER v4.11.1. **(J)** Percentage of peaks lying nearer to the TSS or to the TTS of the nearest gene. From left to right: full list of consensus peaks; peaks with a gain in siDDX5 only; peaks with a gain in both siDDX5 and siXRN2; peaks with a gain in both siDDX5 and PRMT5; and peaks with a gain in siDDX5, siXRN2, and siPRMT5 cells.

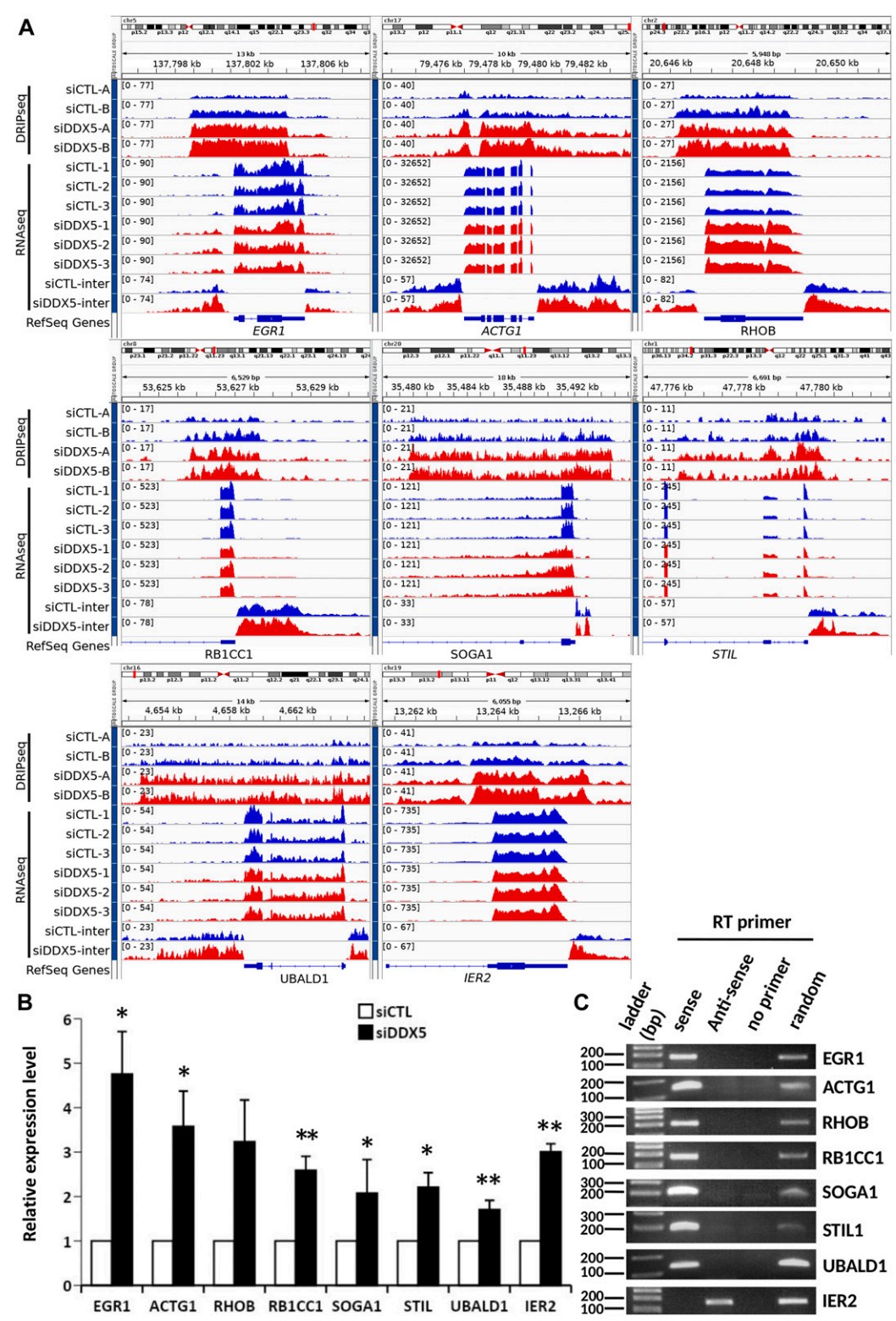

**Figure 5. R-loops in siDDX5 induce antisense transcription.**
**(A)** Read coverage of DRIP-seq and RNA-seq signal centered at the R-loop gain peak associated to the *EGR1*, *ACTG1*, *RHOB*, *RB1CC1*, *SOGA1*, *STIL*, and *UBALD1* genes and the downstream of *IER2* gene loci in siCTL and siDDX5 (absolute $\log_2$ fold change > 1 and false discovery rate < 0.1, Wald test). The bottom two tracks show the intergenic RNA-seq coverage of the pooled replicates. Whereas the gene expression decreased upon the knockdown treatment, the read coverage at the intergenic region adjacent to the transcription start site increased, thus indicating possible antisense transcription. Control cells are blue, siDDX5 cells are red. **(B)** Antisense expression was quantified by reverse transcription quantitative PCR (RT-qPCR) using cDNAs transcribed with random primers. The amplified fragments are located at the peak regions

identified in U2OS cells spanning ~4.5% of the genomic sequence, with R-loop gain peaks being more prevalent in chromosomes with high gene density. The DRIP-seq gain peaks were associated with genomic regions known to be transcriptionally active; however, most DRIP-seq gain peaks did not lie near differentially expressed genes observed by RNA-seq between siCTL and siDDX5, siXRN2, and siPRMT5 cells. These findings confirm that DDX5, XRN2, and PRMT5 are bona fide R-loop regulators, and it is not the differential RNA expression that generates the increased R-loops. In cells deficient for DDX5, XRN2, or PRMT5, an elevated number of peaks with increased R-loop accumulation that overlaid with the TTS was observed, suggesting they function together during transcription termination. In addition, we noted that the intronic reads that had increased accumulation in PRMT5-deficient cells were highly clustered upstream of the TTS. In contrast, intronic reads were equally distributed near the TSS and TTS in DDX5- and XRN2-deficient cells. DDX5-, XRN2-, and PRMT5-deficient cells also had increased reads in the promoter and TSS regions, regions known for their elevated GC-rich sequences or CpG islands, defining a requirement for DDX5, XRN2, and PRMT5 to repress R-loops associated with initiation of gene transcription at certain loci. Finally, we observe that the accumulated R-loops at certain loci in siDDX5, siXRN2, and siPRMT5 cells generated antisense intergenic transcription. Our data provide a genome-wide perspective of R-loops regulated by DDX5, XRN2, and PRMT5.

DDX5 has been shown to function in many cellular RNA processes where structured RNA is implicated and, in addition, DDX5 has been shown to influence gene transcription (Fuller-Pace, 2013; Xing et al, 2017). The role in gene transcription, as suggested using *Drosophila* and yeast DDX5 homologs, was to promote RNA release from chromatin (Buszczak & Spradling, 2006; Cloutier et al, 2012). DDX5 was shown to function as a resolvase for the GC-rich areas forming G-quadruplex of the proximal promoter of *MYC* (Wu et al, 2019). Our data show enrichment of R-loop gains at promoters and TSS regions with high GC-rich content in siDDX5 cells (Fig 4). Actually, an R-loop at the *MYC* locus was increased in DDX5-deficient cells (Supplemental Data 1), consistent with GC-rich regions forming both G-quadruplexes and R-loops. DDX5 associates with the estrogen receptor *α* and is recruited to estrogen-dependent promoters in a cyclic fashion, suggesting DDX5 plays a role in transcription initiation (Endoh et al, 1999; Metivier et al, 2003; Wortham et al, 2009). R-loops were shown to increase co-transcriptionally during estrogen treatment (Stork et al, 2016). Thus, DDX5 is likely required near the TSS to resolve R-loops, allowing transcription and minimize DNA damage at these loci. Estrogen-regulated genes were not enriched in our U2OS data (Supplemental Data 1), as these are likely occurring in a tissue-specific manner in breast cancer cell lines (Stork et al, 2016) and how DDX5 gets recruited at these sites is not known. Importantly, the overexpression of DDX5 in breast cancer has been observed,

validating the therapeutic value of targeting the helicase activity of DDX5 (Mazurek et al, 2012).

The exoribonuclease XRN2 is a known regulator of transcription termination with its "torpedo" function (Kim et al, 2004; Teixeira et al, 2004; West et al, 2004; Eaton & West, 2020). XRN2-deficient cells have replicative stress with increased DNA damage with increased R-loops using S9.6 immunofluorescence staining (Morales et al, 2016). We now define the genome-wide loci where R-loops accumulate in the absence of XRN2. We confirm the classical role of XRN2 at the TTS (Eaton & West, 2020), where the highest R-loop gains were observed. We also noted an increase in R-loop gains near the TSS, consistent with the presence of intergenic RNAs likely to be potential antisense RNAs and whether XRN2 has a role in transcription initiation remains to be determined.

The DRIP-seq results reveal more R-loops affected by PRMT5 deficiency (2,632) than by DDX5 (762) or XRN2 (1,059) deficiency. As PRMT5 posttranslationally modifies proteins, it is not surprising that PRMT5 has an elevated number of cellular R-loops. RNA helicases known to resolve R-loops containing arginine methylation sites (RGG/RG motifs) and potential substrates of PRMT5 include DHX9 (Chakraborty et al, 2018; Cristini et al, 2018) and DDX21 (Song et al, 2017) and there are likely others. These observations suggest PRMT5 may regulate other RNA helicases, besides DDX5, to resolve R-loops. Moreover, PRMT5 methylation of RNA polymerase II C-terminal domain leads to the recruitment of RNA helicase Senataxin to resolve R-loops (Zhao et al, 2016). In addition to RNA helicases, PRMT5 methylates many RNA splicing and processing factors (Guccione & Richard, 2019), where defects in these processes also lead to R-loop accumulation (Li & Manley, 2005; Bhatia et al, 2014). The intronic DRIP-seq reads observed for PRMT5 were highly clustered before the TTS, and these represent nascent RNAs consistent with the role of PRMT5 in pre-mRNA splicing (Friesen et al, 2001; Fong et al, 2019). Thus, it is possible that PRMT5 as well as DDX5 and XRN2 may also influence R-loop accumulation by also affecting the processing and nuclear export of RNAs.

The presence of unscheduled R-loops is associated with increased DNA damage and genomic instability (Skourti-Stathaki & Proudfoot, 2014; Aguilera & Gomez-Gonzalez, 2017; Richard & Manley, 2017; Crossley et al, 2019; Wells et al, 2019). Thus, the presence of elevated number of R-loops gain peaks in siPRMT5 cells is consistent with inhibition of PRMT5 methyltransferase activity causing increased DNA damage as a cancer therapeutic (Clarke et al, 2017; Hamard et al, 2018; Fong et al, 2019; Mersaoui et al, 2019).

Taken together, our data identify a shared role for DDX5, XRN2, and PRMT5 in R-loop metabolism at the TTS region. In addition, we uncovered genomic loci where the loss of DDX5, XRN2, or PRMT5 leads to R-loop–associated antisense expression near the TSS regions. Thus, our data provide a valuable resource of the genome-wide R-loops modulated by DDX5, XRN2, and PRMT5 using classical DRIP-seq.

---

upstream of *EGR1*, *ACTG1*, *RHOB*, *RB1CC1*, *SOGA1*, *STIL*, and *UBALD1* genes and the downstream of *IER2* gene. The relative expression was normalized with *GAPDH*. The graph shows the average and SEM from four independent experiments. Statistical significance was assessed using *t* test. *P < 0.05, ** P < 0.01, and ***P < 0.001. **(C)** Agarose gel image of the RT-PCR products amplified at the promoter region of the *EGR1*, *ACTG1*, *RHOB*, *RB1CC1*, *SOGA1*, *STIL1*, and *UBALD1* genes and downstream of *IER2* gene. The RT primers used are sense primers, which hybrid with the antisense strand corresponding to the gene; antisense primer, annealing to the sense strand; random primers; and no primer. DNA markers are shown on the left are in base pairs.

# Materials and Methods

### Cell culture, siRNAs, and transfection

Human osteosarcoma cells U2OS cells (ATCC) were cultured in Dulbecco's modified Eagle's medium containing 10% vol/vol FBS at 37°C with 5% $CO_2$. siRNA oligonucleotides were transfected using Lipofectamine RNAiMAX (Invitrogen) according to the manufacturer's instructions. All siRNAs were purchased from Dharmacon. siRNA sequences are as follows: siDDX5: 5′-ACA UAA AGC AAG UGA GCG AdTdT-3′; siXRN2, SMARTpool siGENOME human XRN2 siRNA (M-017622-01); siPRMT5, 5′-UGG CAC AAC UUC CGG ACU UUU-3′. The siRNA 5′-CGU ACG CGG AAU ACU UCG AdTdT-3′, targeting the firefly luciferase (GL2), was used as control. 20 nM siRNA was used for transfection.

### DRIP procedure

DRIP assays were performed as described (Ginno et al, 2012). Briefly, nucleic acids were extracted from U2OS cells by SDS/proteinase K treatment at 37°C overnight followed by phenol–chloroform extraction using MaXtractTM High Density (100 × 15 ml from QIAGEN) and ethanol precipitation at room temperature. The harvested nucleic acids were digested for 24 h at 37°C using a restriction enzyme cocktail (50 units/100 µg nucleic acids, each of *Bsr*GI, *Eco*RI, *Hind*III, *Ssp*I, and *Xba*I) in the New England Biolabs CutSmart buffer with 2 mM Spermidine and 1× BSA. Digested DNAs were cleaned up by phenol–chloroform extraction using MaXtractTM High Density (200 × 2 ml) followed by treatment or not with RNase H (20 units/100 µg nucleic acids) overnight at 37°C in the New England Biolabs RNase H buffer. DNA/RNA hybrids from 4.4 µg digested nucleic acids, treated or not with RNase H, were immunoprecipitated using 10 µg of S9.6 antibody (ATCC) and 50 µl of protein A/G agarose beads (Sigma-Aldrich) at 4°C for 2 h or overnight in IP buffer (10 mM $NaPO_4$, 140 mM NaCl, and 0.05% Triton X-100). The beads were then washed four times with IP buffer for 10 min at room temperature, and the nucleic acids were eluted with elution buffer (50 mM Tris–HCl, pH 8.0, 10 mM EDTA, 0.5% SDS, and 70 µg of protease K) at 55°C for 1 h. Immunoprecipitated DNA was then cleaned up by a phenol–chloroform extraction followed by ethanol precipitation at 20°C for 1 h. For each replicate and condition, three IP were combined and sent to IGM genome center of the University of Californa San Diego for library construction and sequencing. Data were generated at the UC San Diego IGM Genomics Center using an Illumina NovaSeq 6000.

### DRIP-seq analysis

Single-end reads of length 76 were first trimmed with fastq-mcf (Aronesty, 2013) to remove the adapter sequence, then mapped to the human genome (hg19/GRCh37) through Bowtie2 (Langmead & Salzberg, 2012), and the resulting Sequence Alignment Map (SAM) files were processed using SAMtools v1.9 (Li et al, 2009) to generate sorted Binary Sequence Alignment Map (BAM) files with duplicate reads removed. Peaks were called for each replicate relative to the input samples through the MACS algorithm (Zhang et al, 2008) in broad mode at a q-value cutoff of 0.1. The resulting peaks were merged into a single list of consensus peaks using the DiffBind R package (Ross-Innes et al, 2012) and the differentially bound sites of both gains and losses were then identified by DESeq2 v1.26.0 at a false discovery rate (FDR) cutoff of 0.1 (Wald test) and absolute log-fold-change larger than one. Consensus peaks were annotated with the nearest and neighboring genes (definition of nearest and neighbor: nearest is defined as the gene whose TSS or TTS lies nearest to the peak, and neighbor is defined as the second nearest gene) by BEDTools v2.28.0 (Quinlan & Hall, 2010) based on the Ensembl gene database obtained from the University of Californa Santa Cruz table browser (Karolchik, 2004), considering only the longest transcript of each gene. Overlapping sets of peaks as well as peak locations within the gene body and GC content histograms were made through HOMER v4.11.1 (Heinz et al, 2010) Enrichment plots of R-loop signal across the gene bodies were generated with ngs.plot v2.63 (Shen et al, 2014). Pathway enrichment analysis of genes lying in the nearest consensus peaks with gain in R-loop signal upon knockdown was made through the Database for Annotation, Visualization, and Integrated Discovery (DAVID) (Huang et al, 2009). In 2016, Sanz et al (2016) carried out a DRIP-seq screen on NT2 cells using the same enzyme as in this study (Sanz et al, 2016). After assigning the nearest gene to each peak of both studies, we found an overlap of 10,786 genes.

### cDNA library preparation and next generation sequencing

Total cellular RNA was isolated from siRNA-transfected U2OS cells using GenElute Mammalian total RNA Miniprep Kit (Sigma-Aldrich) according to the manufacturer's instructions. For each experimental condition, three independent siRNA transfection experiments were carried out for the isolation of RNA. NEB rRNA-depleted (HMR) stranded library preparation was performed using Illumina's platform following the manufacturer's protocol (New England BioLabs). Purified libraries were subjected to sequencing on Illumina HiSeq4000 PE100 in pools of five per lane. Sequencing produced an average of 79.2 million reads per sample (range 71.1–87.7 million).

### RNA-seq analysis

Paired-end reads of length 100 were aligned to the human genome (hg19/GRCh37) with STAR v2.7.1a (Dobin et al, 2013) and the SAM/BAM files were processed using SAMtools v1.9 (Li et al, 2009). Quantification of gene expression for each replicate was performed through HOMER v4.11.1 (Heinz et al, 2010) using Gencode v19 gene annotations (Frankish et al, 2019). The read counts were then normalized across all samples and their differential expression relative to the control computed through DESeq2 v1.26.0 (Love et al, 2014). Up-regulated and down-regulated genes were called at an absolute $\log_2$ fold change larger than one and FDR cutoff of 0.05, Wald test. Overlaps between differentially expressed genes and genes lying nearest gain or loss R-loop peaks were determined by HOMER. To detect cases of antisense transcription, the intergenic reads of each replicates were extracted with BEDTools v2.28.0 (Quinlan & Hall, 2010) using the University of Californa Santa Cruz table browser database of Ensembl genes (Karolchik, 2004).

**Table 2. Primers for reverse transcription (RT)-PCR analysis.**

| Gene | Primer |
|---|---|
| EGR1 | RT sense: 5′-CCCTGTTCGCGTTCGGCCCC-3′ |
| | RT antisense: 5′-GCTCGGTGCTGCCCCCTGGAG-3′ |
| | PCR forward: 5′-CACCCCCTGCTTCCTTCTCC-3′ |
| | PCR reverse: 5′-CGACGCAGTGAGCACGAACT-3′ |
| ACTG1 | RT sense: 5′-CGGAGCAGAACGTAG-3′ |
| | RT antisense: 5′-GCCCAGAATCTCCGG-3′ |
| | PCR forward: 5′-GTGTCCCTCGGTGTGTGACG-3′ |
| | PCR reverse: 5′-CGGGCAAGGCTGTCAGGTAT-3′ |
| RHOB | RT sense: 5′-CGGGACTTGGAAGAG-3′ |
| | RT antisense: 5′-GCTCTGGCGGTACCC-3′ |
| | PCR forward: 5′-GGGGCCCTAAACCACAGGAG-3′ |
| | PCR reverse: 5′-GCCCCTCTTCCTGGCAAACT-3′ |
| RB1CC1 | RT sense: 5′-CGGGACTTGGAAGAG-3′ |
| | RT antisense: 5′-GCTTGTTCCCCTCAG-3′ |
| | PCR forward: 5′-TCCCAACCATTAGGGTGCTCA-3′ |
| | PCR reverse: 5′-GCGGCACCATTTCTCAGACC-3′ |
| SOGA1 | RT sense: 5′-GAGATGGAGTCTAGC-3′ |
| | RT antisense: 5′-CAGGAGTTCGAGACC-3′ |
| | PCR forward: 5′-ACCTCGGCTCACTGCAACCT-3′ |
| | PCR reverse: 5′-CCAACATGATGAAACCCCGTCT-3′ |
| STIL | RT sense: 5′-TTGAACTCGGGAGGC-3′ |
| | RT antisense: 5′-CGCGCTCGACCAATC-3′ |
| | PCR forward: 5′-GTTCTTCGGGTGTCCGCTTC-3′ |
| | PCR reverse: 5′-CGGCGCTCCAGGATCAAG-3′ |
| UBALD1 | RT sense: 5′-TAGAGACGGTTTGAC-3′ |
| | RT antisense: 5′-TTCCTGGCCCTGACC-3′ |
| | PCR forward: 5′-GTCCTGGGCCTAGGCAATCC-3′ |
| | PCR reverse: 5′-GGGAGCGAATTTCGGAAACC-3′ |
| IER2 | RT sense: 5′-CCGGTTACCACGTGG-3′ |
| | RT antisense: 5′-TGATACTGTAGGGCC-3′ |
| | PCR forward: 5′-CGGGCATTCCCTAACTGGTG-3′ |
| | PCR reverse: 5′-GTGCAATCGATCCCCAGCTC-3′ |

Differential expression at the intergenic regions within 5 kbp of either the TSS or TTS of the nearest gene for each DRIP-seq peak relative to the control samples was then evaluated through HOMER. The resulting hits were filtered using HOMER to lie within 5 kbp of the nearest R-loop consensus gain peak, ensuring as well that the expression of the nearest gene is not up-regulated relative to the control. Antisense hits reported in Supplemental Data 3 were selected through the following thresholds: (1) a positive $\log_2$ fold change of differential intergenic RNAseq expression relative to the control at the 2 kb adjacent to the TSS or TTS, (2) $\log_2$ fold change of differential gene expression relative to the control smaller than 2, (3) $\log_2$ fold change of R-loop peak signal relative to the control larger than one, and (4) FDR of R-loop peak signal relative to the control smaller than 0.1.

**Table 3. Primers pairs for antisense RT-qPCR analysis.**

| Gene | Primer |
|---|---|
| EGR1 | 5′-AGGCTCGGGGTGAGGAGTGT-3′ |
| | 5′-CGACGCAGTGAGCACGAACT-3′ |
| ACTG1 | 5′-GTGTCCCTCGGTGTGTGACG-3′ |
| | 5′-CAACAGACCCACCCGGACTC-3′ |
| RHOB | 5′-GCCAGGAAGAGGGGCAATTC-3′ |
| | 5′-GTCCGGGAGCTGGCTGTCT-3′ |
| RB1CC1 | 5′-TCCCAACCATTAGGGTGCTCA-3′ |
| | 5′-CGCCACAACCACGTTTTCAG-3′ |
| SOGA1 | 5′-ACCTCGGCTCACTGCAACCT-3′ |
| | 5′-CAAATTAGCCGGGCGTGGTA-3′ |
| STIL | 5′-GTTCTTCGGGTGTCCGCTTC-3′ |
| | 5′-CGCAATGGAAAGCCCAGCTA-3′ |
| UBALD1 | 5′-TCCTCGGACCCCGAGTAGGT-3′ |
| | 5′-GGGAGCGAATTTCGGAAACC-3′ |
| IER2 | 5′-CGGGCATTCCCTAACTGGTG-3′ |
| | 5′-AAAGCCCCGATCTCCCTGTC-3′ |

## RT and quantitative PCR (RT-qPCR) analysis for antisense RNA expression

For antisense RNA analysis, total cellular RNA was isolated as described above. cDNA was synthetized using Promega M-MLV reverse transcription kit. For determining which strand of RNA was transcribed, two different primers were used in the RT reaction. The sense primer hybrids with the antisense strand and the antisense primer anneals to the sense strand. A random primer was used as

**Table 4. Primers for DRIP-qPCR validation.**

| Gene | Primer |
|---|---|
| FOS | 5′-CCTGCAAGATCCCTGATGACCT-3′ |
| | 5′-AGGGTGAAGGCCTCCTCAGACT-3′ |
| KLF2 | 5′-GACAACAGTGGGGAGTGGACCTT-3′ |
| | 5′-CTGAGGGATCCTTGCCCTACATC-3′ |
| JUNB | 5′-CCGGATGTGCACTAAAATGGAAC-3′ |
| | 5′-AGTCGTGTAGAGAGAGGCCACCA-3′ |
| CTNNB1 | 5′-GCCATTTTAAGCCTCTCGGTCTG-3′ |
| | 5′-CTCCTCAGACCTTCCTCCGTCTC-3′ |
| LY6E | 5′-GAAGGCTGCTGAGTTTCCTCCTC-3′ |
| | 5′-GCTTCTCTCCTGACCCACTCCTC-3′ |
| SNHG12 | 5′-CTGGGACTATAAGCACGCACCAC-3′ |
| | 5′-TTGGGGTCAGGAGTTCAAGACTG-3′ |
| SOWAHC | 5′-GCTAGCCTTCTGGGAAAAGTGGA-3′ |
| | 5′-GAAGTGGAGGGCAGAGAAGAGGT-3′ |
| RPS23-1 | 5′-TTAGTCGGTTCAGGGCAACTTGA-3′ |
| | 5′-CTAAGACACTCGCCTCACCTGGA-3′ |
| RPS23-2 | 5′-GTTCATGCCTGTAATCCCAGCAC-3′ |
| | 5′-GTATGACTTTGCTGCCCAGGATG-3′ |

positive control and for the negative control, no primer was added in the RT reaction. Regular PCR reaction was performed using the cDNA transcribed with different primers, and the PCR product was revealed in agarose gel. Antisense expression was quantified by RT-qPCR using SYBR Green PCR Master Mix (Applied Biosystems) and cDNAs transcribed using random primers. The sequence of the primers for RT reaction and PCR to identify sense or antisense transcription are listed in Table2 and the primers for qPCR are in Table 3. *Gapdh* housekeeping gene expression was used to normalize antisense expression. DRIP-qPCR analysis was performed as described previously (Mersaoui et al, 2019). The primers are listed in Table 4.

# Supplementary Information

# Acknowledgements

This work was funded by FDN-154303 to S Richard and FDN-388879 to J-Y Masson. SY Mersaoui holds a Fonds de recherche du Québec-Santé (FRQS) fellowship and J-Y Masson is a Tier I Canada Research Chair in DNA repair and cancer therapeutics.

## Author Contribution

OD Villarreal: conceptualization, resources, data curation, software, formal analysis, methodology, project administration, and writing—original draft, review, and editing.
SY Mersaoui: data curation, formal analysis, methodology, and writing—review and editing.
Z Yu: conceptualization, data curation, formal analysis, supervision, methodology, writing—original draft, and project administration.
J-Y Masson: conceptualization, supervision, funding acquisition, project administration, and writing—review and editing.
S Richard: conceptualization, supervision, funding acquisition, project administration, and writing—original draft, review, and editing.

## Conflict of Interest Statement

The authors declare that they have no conflict of interest.

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
