## [Reviewer comments · Life Science Alliance]

Life Science Alliance

Genome-wide Rloop analysis defines unique roles for DDX5, XRN2 and PRMT5 in DNA/RNA hybrid resolution

Oscar Villarreal, Sofiane Mersaoui, Zhenbao Yu, Jean-Yves Masson, and Stéphane Richard
DOI: <https://doi.org/10.26508/lsa.202000762>

Corresponding author(s): Stéphane Richard, McGill University

Review Timeline:	Submission Date:	2020-05-01
	Editorial Decision:	2020-06-05
	Revision Received:	2020-07-08
	Editorial Decision:	2020-07-22
	Revision Received:	2020-07-23
	Accepted:	2020-07-24

Transaction Report:

June 5, 2020

Re: Life Science Alliance manuscript #LSA-2020-00762

Prof. Stephane Richard
McGill University
Departments of Oncology and Medicine
Lady Davis Institute for Medical Research
3755 Cote-Ste-Catherine Road
Montreal, Quebec H3T 1E2
Canada

Dear Dr. Richard,

Thank you for submitting your manuscript entitled "Genome-wide R-loop analysis defines roles for DDX5, XRN2 and PRMT5 in DNA/RNA hybrid resolution." to Life Science Alliance. The manuscript was assessed by expert reviewers, whose comments are appended to this letter.

While all three reviewers see the systematic analysis of R-loop formation in cells with reduced DDX5, PRMT5 or XRN2 levels as valuable, they expect additional support for the conclusions drawn as well as further insights into the potential mechanisms underlying the effects you report. Both Reviewers provide constructive input on how to provide such support and insight. We think the requests are reasonable and addressable, and would thus like to invite you to submit a revised version addressing all the reviewers comments thoroughly.

Thank you for this interesting contribution to Life Science Alliance. We are looking forward to receiving your revised manuscript.

Sincerely,

Reilly Lorenz
Editorial Office Life Science Alliance
Meyerhofstr. 1
69117 Heidelberg, Germany
t +49 6221 8891 414
e contact@life-science-alliance.org
www.life-science-alliance.org

B. MANUSCRIPT ORGANIZATION AND FORMATTING:

Reviewer #1 (Comments to the Authors (Required)):

Mersaoui et al "Genome-wide R-loop analysis defines unique roles for DDX5, XRN2 and PRMT5 in DNA/RNA hybrid resolution"

The manuscript characterizes R-loop formation on a genome-wide scale at transcription termination sites upon depletion of DDX5, XRN2 and PRMT5, three key R-loop-resolving factors. The authors IP DNA/RNA hybrids coupled to HT sequencing (DRIP-seq) to show that these factors regulate R-loops at TTSs in both unique and overlapping ways at key gene groups. DDX5, in contrast with the other factors, also plays a role in R-loop formation at TSSs. Notably, at these sites, DDX5-depletion and R-loop formation induces anti-sense transcription. Finally, gene ontology analysis demonstrated that PRMT5 uniquely regulates TTSs of chromatin-modifying genes.

Overall this is an important study that illuminates the role of key R-loop regulatory factors at a genome-wide level and reveals overlapping and specific pathways in gene regulation.

Specific comments:

The genome-wide analysis is the strength of this manuscript. However, it would be interesting to reconstitute one of the knockdowns with one of the other factors to see if this could complement the loss at an overlapping knockdown-induced R-loop. These experiments may be beyond the scope of the manuscript but then discussing this idea could be helpful.

Reviewer #2 (Comments to the Authors (Required)):

The authors identify DDX5, XRN2, and PRMT5 as R-loop regulators and thoroughly characterize the changes in R-loop levels in U2OS cells that are deficient for DDX5, XRN2, and PRMT5 using genome-wide DRIP-seq. As these genes are individually silenced, R-loops appear to be exacerbated independent of the expression levels of nearby genes. Moreover, cells that are deficient for DDX5 are shown to have an accompanying increase in antisense transcription in loci that also have an increased R-loop signal. Despite the lack of functional significance of these antisense RNAs, this is an interesting observation that lends support to a new publication that showed R-loops may act as promoters for RNA pol II-dependent transcription and facilitate antisense transcription. Gained R-loops in the individual knockdown lines are associated with different GO pathways, but the changes in R-loops are not accompanied by changes in transcription, and it's not known how the regulation of R-loops affects the genes in the various GO pathways.

MAJOR:

1) Introduction: There is extensive background on proteins involved in R-loop regulation but little background on PRMT5. Shifting the focus of the introduction more towards PRMT5, DDX5 and XRN2 will be useful such that previous literature on the functional relationship i) among these proteins, for example, protein-protein interactions, and ii) between these genes and R-loops is discussed more thoroughly. Moreover, the rationale for why it is interesting to study the three proteins further is not clear.

2) Page 7: R-loop peaks are called relative to input and it is stated that the "majority of the peaks are RNase H sensitive". The paper would greatly benefit from a more quantitative analysis related to the proportion of peaks that are RNase H sensitive and the level of sensitivity (i.e. fold-change in peak coverage between control and RNase H).

- 3) Figure 3A: The statistical tests to measure if distances are significantly different are missing.
- 4) "...the R-loop gain peaks at the 5'-terminus (promoter-TSS and 5'-UTR) showed an increase only in siDDX5-deficient cells (13.5% + 1.2%), but not in the XRN2-deficient (9.9% + 0.5%) or PRMT5-deficient cells (5.7% + 0.5%) compared to control cells (8.2% + 1.0%, Figure 3A)"
- This should be referring to Figure 4A and not Figure 3A.
 - For XRN2-deficient cells, the numbers according to the pie chart are 9.9% + 0.9% and not 9.9% + 0.5%. Please correct the discrepancy.
 - 9.9%+0.9% in siXRN2 is also larger than 8.2%+1% in control cells. What do you consider a difference vs. no difference here? Are these significant or not significant differences?
- 5) Figure 4C "In cells deficient for DDX5, XRN2 or PRMT5, we observed a higher number of peak reads that overlaid with the TSS". There seems to be a higher number of reads overlapping TSS and gene bodies as well. Based on these results, how can the authors be confident that DDX5, XRN2, and PRMT5 function together in transcription termination, and not at transcription start sites or gene bodies?
- 6) Did the authors consider the possibility that the targeted proteins may be preventing the accumulation of R-loops rather than resolving them? Please clarify/discuss.
- 7) Do the authors observe an increase in antisense transcription in siXRN2 and siPRMT5 cells, similar to siDDX5 cells? If R-loops act as promoters for antisense Pol II transcription, there should also be an increase in antisense transcription in siXRN2 and siPRMT5 cells.
- 8) In the methods section and Figure 5C legend the authors state "Antisense expression was quantified by RT-qPCR using SYBR Green PCR Master Mix (Applied Biosystems) and cDNAs transcribed using random primers" and "Antisense expression was quantified by real-time quantitative PCR (RT-PCR) using cDNAs transcribed with random primers". If random primers are used, how can antisense expression be specifically measured vs. sense strand? Is the RNA-seq data stranded? If not, how can the authors conclude that the reads are antisense and not sense? For Figure 5A, the authors state "While the gene expression decreased upon DDX5 depletion, the read coverage at the intergenic region adjacent to the TSS increased, thus indicating antisense transcription". If this is how antisense transcription is inferred genome-wide, the data would be less convincing. Please clarify.
- 9) On page 14 the authors state, "Gene ontology analysis of R-loop peaks identified specific categories of genes regulated by DDX5, XRN2 and PRMT5 such as those involved in ATP binding, DNA damage signaling and histone modifications, respectively". Since the changes in R-loops are not accompanied by changes in transcription as determined by RNA-seq, what would the mechanism of R-loop-mediated gene regulation be in these cases? How do the authors think PRMT5, DDX5, and XRN2 regulate genes in these GO pathways in a way that is dependent on R-loops but independent of mRNA level?

MINOR:

- 1) Throughout the results section, the appearance of some figure panels is not in chronological order. Figure 1C is discussed before Figure 1B and similarly Figure 6D appears in the text before Figure 6B-C. If the text or figures are reorganized, the text will have a better flow. Figures 1D and E belong with Figure 2A-C where gain/loss of R-loop peaks are demonstrated. The

authors should consider revising the organization so that Figure 1 has three panels (A-C) and Figure 2 has five panels (A-E).

2) Figure S3E- For the Venn diagrams, keep the order of the different knock-down lines the same as in the upper panel (DDX5, XRN2, and PRMT5) for consistency.

3) The methods section is missing the company from which the S9.6 antibody and protein A/G beads are purchased.

4) The EGR1 genome track is shown twice in Figure 5 and Figure S5. These appear to be identical.

5) The rationale for analyzing "intronic reads" is not clear. Are these representing nascent RNA molecules? A short explanation in the text would be useful.

6) The RefSeq tracks in IGV appear to be "collapsed" where overlapping annotations are shown on top of one another. The track can easily be "expanded" in IGV which would allow for better visualization of the TSS and TTS for genes that are overlapping such as the ones in Figure S1B (SSTR5-AS1 and SSTR5). As is, it is hard to distinguish the two ORFs.

7) The word "significant" is used without implying statistical significance throughout the manuscript i.e pg 12. Please replace these with other adjectives that don't imply statistical significance if the statistical test is not shown as in Figure 5A and S5A.

8) Page 8: "Less significant overlap of loss peaks". Since the p-values are extremely low, these differences are certainly still very significant and it should not be referred to as "less significant".

9) If the cutoffs for fold change and FDR are marked on the volcano plots in Figure 1D and Figure S3C, it would be easier to interpret the results. As is, the cutoffs are only mentioned in the materials section and harder to access.

10) Figure S1A: The legend states the scatter plots represent RNase H treated samples. Is this a typo?

Reviewer #3 (Comments to the Authors (Required)):

The article by the Richard lab maps R-loops in the genome of cells siRNA depleted for DDX5, PRMT5 and XRN2. This builds on previous work showing that these factors regulate R-loop levels and is a new effort to identify genomic features associated with R-loops for each factor. The data support the model that DDX5, PRMT5 and XRN2 work together to regulate R-loops at some transcription termination sites (TTS). In addition, a new, possibly independent role for DDX5 in regulating antisense transcription at promoters was proposed based on the changes in DRIP profiles for siDDX5 cells. The manuscript is straightforward and primarily a descriptive analysis of DRIP-seq data. Investigation and validation of the new DDX5 role at promoters is limited to a set of RT-PCR experiments but hopefully will stimulate research in the field. Below are some suggestions to improve the article.

Comments:

- Despite the significant overlap of peaks the differences between DDX5, XRN2 and PRMT5 seem to be an important new finding in this study. PRMT5 knockdown seems to elicit way more new

peaks than either of the other two knockdowns (true for both gained and lost peaks). Is there a technical or biological reason for this? For example, 345 DDX5 peaks overlap with PRMT5, which leaves >2000 PRMT5 peaks that are non-overlapping! The discussion seems to suggest a biological rationale of a splicing defect. Is such a splicing defect evident in the RNA-seq data? Part of this discussion could also include the rationale for focussing on DDX5 as opposed to PRMT5 (which seems to cause a stronger effect on the genome).

- Some of the numbers do not appear to match or are presented in a confusing way. For example, on page 8, 762 DDX5 peaks are identified, but in the comparison to RNA-seq on Page 9 it says "...of the siDDX5 697 R-loop gain peaks...". Which 697? Is this a subset of the 762 or a different set?

- The siPRMT5 induced R-loops did not appear closer to neighboring genes. Is this a statistical artifact because there are so many more PRMT5 peaks?

- Confirming by RT-PCR the changes in antisense transcription is an important result. Is there any data to suggest that DDX5 is physically associated to these TSS in control cells? Do the authors have any insight to how DDX5 is recruited? ChIP experiments with DDX5 could help to answer these questions.

- Have the authors considered analysis promoter-proximal pause sites? I am not sure if a pausing index dataset is available in U2OS cells but since DDX5 binds to POLII, it may present an opportunity to give more insight into the regulation of spurious antisense transcription in cells lacking DDX5.

- I was not convinced by the GO analysis in Figure 6. The formation of R-loops is governed by genomic position, GC content, transcription frequency, etc. Its not clear to me that the changes in R-loop position will have functional consequences for the target genes identified or if the enrichments are basically coincidental because of physical features of the genes. There also does not appear to be a consistent change in gene expression (Fig. 6D) based on the R-loop gains. The discussion states that "DDX5 is likely required near the TSS to resolve R-loops, allowing transcription..." - if true then the shouldn't DDX5 R-loop gained genes be repressed?

Minor points:

The original reference for the AQR helicase in the introduction is missing (Sollier et al., Mol Cell. 2014).

Reviewer #1 (Comments to the Authors (Required)):

Mersaoui et al "Genome-wide R-loop analysis defines unique roles for DDX5, XRN2 and PRMT5 in DNA/RNA hybrid resolution"

The manuscript characterizes R-loop formation on a genome-wide scale at transcription termination sites upon depletion of DDX5, XRN2 and PRMT5, three key R-loop-resolving factors. The authors IP DNA/RNA hybrids coupled to HT sequencing (DRIP-seq) to show that these factors regulate R-loops at TTSs in both unique and overlapping ways at key gene groups. DDX5, in contrast with the other factors, also plays a role in R-loop formation at TSSs. Notably, at these sites, DDX5-depletion and R-loop formation induces anti-sense transcription. Finally, gene ontology analysis demonstrated that PRMT5 uniquely regulates TTSs of chromatin-modifying genes.

Overall this is an important study that illuminates the role of key R-loop regulatory factors at a genome-wide level and reveals overlapping and specific pathways in gene regulation.

Specific comments:

The genome-wide analysis is the strength of this manuscript. However, it would be interesting to reconstitute one of the knockdowns with one of the other factors to see if this could complement the loss at an overlapping knockdown-induced R-loop. These experiments may be beyond the scope of the manuscript but then discussing this idea could be helpful.

REPLY: The reconstitution experiments especially using various mutant variants of the enzymes would be interesting. Controlling the proper ectopic expression of the enzymes to mimic endogenous levels is technically challenging and for this reason we think it is beyond the scope of our current manuscript.

Reviewer #2 (Comments to the Authors (Required)):

The authors identify DDX5, XRN2, and PRMT5 as R-loop regulators and thoroughly characterize the changes in R-loop levels in U2OS cells that are deficient for DDX5, XRN2, and PRMT5 using genome-wide DRIP-seq. As these genes are individually silenced, R-loops appear to be exacerbated independent of the expression levels of nearby genes. Moreover, cells that are deficient for DDX5 are shown to have an accompanying increase in antisense transcription in loci that also have an increased R-loop signal. Despite the lack of functional significance of these antisense RNAs, this is an interesting observation that lends support to a new publication that showed R-loops may act as promoters for RNA pol II-dependent transcription and facilitate antisense transcription. Gained R-loops in the individual knockdown lines are associated with different GO pathways, but the changes in R-loops are not accompanied by changes in transcription, and it's not known how the regulation of R-loops affects the genes in the various GO pathways.

MAJOR:

1) Introduction: There is extensive background on proteins involved in R-loop regulation but little background on PRMT5. Shifting the focus of the introduction more towards PRMT5, DDX5 and XRN2 will be useful such that previous literature on the functional relationship i) among these proteins, for example, protein-protein interactions, and ii) between these genes and R-loops is discussed more thoroughly. Moreover, the rationale for why it is interesting to study the three proteins further is not clear.

REPLY: We have edited the introduction to introduce better XRN2 and PRMT5. The following text was added *“In addition, when formed, R-loops can be removed by cellular RNA nucleases including RNase H1 and RNase H2 which specifically degrade the RNA moiety of R-loops (Wahba et al., 2011) and the 5’-3’ exoribonuclease 2 (XRN2) which degrades nascent RNA downstream the 3’-terminal cleavage site to facilitate transcriptional termination. XRN2 physically associates with the helicases Senataxin (Skourti-Stathaki et al., 2011) and DDX5 (Mersaoui et al., 2019). The helicases resolve the DNA-RNA hybrids to allow the RNA degradation by XRN2 (Mersaoui et al., 2019; Skourti-Stathaki et al., 2011). Deficiency of XRN2 expression causes R-loop accumulation and DNA damage (Mersaoui et al., 2019; Morales et al., 2016).”* and *“Accumulating evidence indicates that protein arginine methylation plays an important role in R-loop metabolism (Mersaoui et al., 2019; Yang et al., 2014; Zhao et al., 2016). Protein arginine methylation is catalyzed by a family of nine protein arginine methyltransferases (PRMTs) (Bedford and Clarke, 2009). PRMT5 is a type II enzyme that catalyzes the symmetrical arginine dimethylation of protein substrates (Guccione and Richard, 2019). PRMT5 has multiple cellular functions and its depletion causes aberrant RNA splicing, DNA damage, R-loop accumulation and genomic instability in multiple cell types (Guccione and Richard, 2019; Yang and Bedford, 2013).”*

2) Page 7: R-loop peaks are called relative to input and it is stated that the "majority of the peaks are RNase H sensitive". The paper would greatly benefit from a more quantitative analysis related to the proportion of peaks that are RNase H sensitive and the level of sensitivity (i.e. fold-change in peak coverage between control and RNase H).

REPLY: We added fold change data in peak coverage between control and RNase H to the Dataset S1 (columns AF-AH titled “siCTL_vs_RnaseH_Fold”, “siCTL_vs_RnaseH_p.value”, “siCTL_vs_RnaseH_FDR”) as well as to Figure S1C and S1D (see siCTL vs RNase-H; right panels) to show that the majority of the peaks are RNase H sensitive.

3) Figure 3A: The statistical tests to measure if distances are significantly different are missing.

REPLY: We added *p* values to Figure 3A.

4) “..the R-loop gain peaks at the 5'-terminus (promoter-TSS and 5'-UTR) showed an increase only in siDDX5-deficient cells (13.5% + 1.2%), but not in the XRN2-deficient (9.9% + 0.5%) or PRMT5-deficient cells (5.7% + 0.5%) compared to control cells (8.2% + 1.0%, Figure 3A)”

a) This should be referring to Figure 4A and not Figure 3A.

REPLY: We corrected it to Figure 4A.

b) For XRN2-deficient cells, the numbers according to the pie chart are 9.9% + 0.9% and not 9.9% + 0.5%. Please correct the discrepancy.

REPLY: 0.5% was edited as 0.9%.

c) 9.9%+0.9% in siXRN2 is also larger than 8.2%+1% in control cells. What do you consider a difference vs. no difference here? Are these significant or not significant differences?

REPLY: We clarified that both siDDX5 and siXRN2 have larger percentages of peaks lying at the 5'-terminus than the control cells. We also added figures S4A and S4B in order to show the statistical significance of the differences in the distance from the R-loop peaks to the nearest TSS or TTS, which is in agreement with Figure 4A. We added the following text based on these new figures: *"In comparison to the unchanged peaks (i.e. those without a gain in any condition), the siDDX5 and siXRN2 gain peaks both lied significantly closer to the nearest TSS ($p=2.8e-11$ and $p=1.1e-11$, Student's t -test) or to the TTS ($p=1.5e-11$ and $p<2.22e-16$, Student's t -test, Figure S4A, S4B). PRMT5-deficient gain peaks lied significantly closer to the nearest TTS (Figure S4B, $p<2.22e-16$), but further from TSS (Figure S4A, $p<1.7e-06$)."*

5) Figure 4C "In cells deficient for DDX5, XRN2 or PRMT5, we observed a higher number of peak reads that overlaid with the TTS". There seems to be a higher number of reads overlapping TSS and gene bodies as well. Based on these results, how can the authors be confident that DDX5, XRN2, and PRMT5 function together in transcription termination, and not at transcription start sites or gene bodies?

REPLY: We added new panels to Figure 4 (panels D, F, H) showing the ratio of peak reads subtracted from siCTL to more clearly show that while DDX5, XRN2 and PRMT5 all lead to a higher number of peak reads overlaying TTS, all three DDX5, XRN2 and PRMT5 lead to R-loop accumulation near the TSS (see Figure 4C) and for PRMT5 there is 1067 R-loop gains at the TSS with 428 and 420 for DDX5 and XRN2, respectively (Figure 4I). We have re-written the result section for Figure 4 to expand roles for XRN2 and PRMT5 as well at the TSS. The following text was edited in the results section:

"Examination of the distribution of the peak reads, showed a higher number from the TSS to the TTS in cells deficient for DDX5, XRN2 or PRMT5 compared to siCTL cells (Figure 4C). Subtraction of siCTL reads shows a better distribution of the peak reads of DDX5, XRN2 and PRMT5 samples (Figure 4D). In the case of DDX5 and XRN2, the peaks were mainly concentrated at the TSS and TTS with less peaks in the gene bodies (Figure 4D). Interestingly, for siPRMT5 cells there was a gradual increase in peak reads from the TSS to the TTS (Figure 4D). We next removed all the reads within the gene bodies revealing an increase in intergenic reads upstream and downstream of the TSS and TTS for DDX5, XRN2 and PRMT5 compared to siCTL (Figure 4E) and this distribution was similar for DDX5, XRN2 and PRMT5 when subtracting siCTL reads (Figure 4F). Lastly, we examined only the intronic R-loop peaks for siDDX5, siXRN2 and siPRMT5 cells. The intronic reads for DDX5, XRN2 or PRMT5 depleted cells were highly clustered before the TTS, while intronic reads near the TSS were mainly present in siDDX5 and siXRN2 cells (Figure 4G, 4H). The subtraction of siCTL revealed a striking pattern of intronic reads for siPRMT5 cells being low near the TSS and accumulating to high levels at the TTS (Figure 4H). Overall, the majority of R-loop peak reads was distributed equally in siDDX5, siXRN and siPRMT5 cells, especially at the TTS, consistent with a coordinated role in transcription termination. The unique read peak patterns in siPRMT5 cells, especially the gradual increase in intronic peak reads from the TSS to TTS, suggests PRMT5 has DDX5- and XRN2-independent roles in nascent RNA regulation."

“DDX5, XRN2 and PRMT5-deficient cells regulate R-loop formation to different levels at transcription initiation sites.

*Although the percentage of R-loop gain peaks at the 3'-terminus (3'-UTR + TTS) increased in DDX5, XRN2 and PRMT5-deficient cells, the R-loop gain peaks at the 5'-terminus (promoter-TSS and 5'-UTR) relative to the unchanged peaks (8.2% +1.0%, Figure 4A) showed an increase in siDDX5-deficient (13.5% + 1.2%), and XRN2-deficient cells (9.9% + 0.9%) and was lower in PRMT5-deficient cells (5.7% + 0.5%, Figure 4A). In fact, PRMT5-deficient cells had 1067 absolute R-loop gains at the TSS, while DDX5-deficient and XRN2-deficient cells had 428 and 420, respectively (Figure 4I). Thus, it is the elevated peak reads at the TTS for PRMT5-deficient cells (Figure 4D) that gives the apparent lower percentage in Figure 4A at the TSS. We then divided the R-loop peaks into two groups in which the center of a R-loop is closer to nearest TSS or TTS. In comparison to the unchanged peaks (i.e. those without a gain in any condition), the siDDX5 and siXRN2 gain peaks both lied significantly closer to the nearest TSS ($p=2.8e-11$ and $p=1.1e-11$, Student's *t*-test) or to the TTS ($p=1.5e-11$ and $p<2.22e-16$, Student's *t*-test, Figure S4A, S4B). PRMT5-deficient gain peaks lied significantly closer to the nearest TTS (Figure S4B, $p<2.22e-16$), but further from TSS (Figure S4A, $p<1.7e-06$). Importantly, the siDDX5 gain peaks were more likely to be located closer to the TSS (56%) than to the TTS (44%), while the siXRN2 and siPRMT5 gains behaved in an opposite manner (40% and 41% near TSS and 60% and 59% near TTS, respectively, Figure 4I). Of the 428 DDX5 R-loop gain peaks near the TSS, we distributed them into 10 groups (Figure 4J); all peaks at the TSS and TTS; R-loop gain peaks from siDDX5 cells alone near the TSS or TTS; R-loop gain peaks from siDDX5 that overlap with R-loop gain peaks from siXRN2 cells near the TSS or TTS; R-loop gain peaks from siDDX5 that overlap with R-loop gain peaks from siPRMT5 cells near the TSS or TTS; R-loop gain peaks from siDDX5 that overlap with R-loop gain peaks from siXRN2 and R-loop gain peaks from siPRMT5 cells near the TSS or TTS. The group with the highest percentage (~75%) was siDDX5 cells alone near the TSS, suggesting a role for DDX5 at the TSS independent of XRN2 and PRMT5. The other group with the highest percentage was siDDX5, siXRN2 and siPRMT5 at the TTS (Figure 4J), further indicating a coordinated role in transcription termination.*

Inspection of the nucleotide sequence of the R-loops generated by siDDX5, siXRN2 or siPRMT5 gain peaks show an increased GC rich content compared to all peaks (Figure S4C, S4D). These observations are consistent with the siDDX5 R-loop gain peaks enriched at TSS and being rich in CG rich areas (CpG islands; S4D), known to be symmetrically distributed near the TSS of promoters (Saxonov et al., 2006). These observations implicate a role for the R-loops generated by DDX5, XRN2 and PRMT5-deficiency in transcription regulation of CpG island genes.”

6) Did the authors consider the possibility that the targeted proteins may be preventing the accumulation of R-loops rather than resolving them? Please clarify/discuss.

REPLY: It is known that splicing factors and RNA export machinery play a role in preventing the accumulation of R-loops rather than resolving them *per se*. We have modified our discussion to include this possibility. “In addition to RNA helicases, PRMT5 methylates many RNA splicing and processing factors (Guccione and Richard, 2019), where defects in these processes also lead to R-loop accumulation (Bhatia et al., 2014; Li and Manley, 2005). The intronic DRIP-seq reads observed for PRMT5 were highly clustered before the TTS and these represent nascent RNAs consistent with the role of PRMT5 in pre-mRNA splicing (Fong et al., 2019; Friesen et al.,

2001). Thus it is possible that PRMT5 as well as DDX5 and XRN2 may also influence R-loop accumulation by also affecting the processing and nuclear export of RNAs.”

7) Do the authors observe an increase in antisense transcription in siXRN2 and siPRMT5 cells, similar to siDDX5 cells? If R-loops act as promoters for antisense Pol II transcription, there should also be an increase in antisense transcription in siXRN2 and siPRMT5 cells.

REPLY: The reviewer raises an important point. Indeed we observed antisense Pol II transcription in siXRN2 and siPRMT5 cells (see Dataset S3 and Figure S5).

8) In the methods section and Figure 5C legend the authors state "Antisense expression was quantified by RT-qPCR using SYBR Green PCR Master Mix (Applied Biosystems) and cDNAs transcribed using random primers" and "Antisense expression was quantified by real-time quantitative PCR (RT-PCR) using cDNAs transcribed with random primers". If random primers are used, how can antisense expression be specifically measured vs. sense strand? Is the RNA-seq data stranded? If not, how can the authors conclude that the reads are antisense and not sense? For Figure 5A, the authors state "While the gene expression decreased upon DDX5 depletion, the read coverage at the intergenic region adjacent to the TSS increased, thus indicating antisense transcription". If this is how antisense transcription is inferred genome-wide, the data would be less convincing. Please clarify.

REPLY: In our previous version, we only confirmed that the transcribed strand upstream of the *EGR1* gene was antisense using strand-specific primers in the reverse transcription (RT) reactions (old Figure 5B). We now provide confirmation of antisense transcription upstream of *EGR1*, *ACTG1*, *RHOB*, *RB1CC1*, *SOGA1*, *STIL* and *UBALD1* loci (Figure 5C). For the quantification analysis (RT-qPCR), we used the GAPDH reference gene to normalize the data and for this we used random primers for real time PCR (Figure 5B). We also used random primers to prime the RT as controls in Figure 5C.

Our RNA-seq was not strand-specific. We have softened our discussions to read “*While the gene expression decreased upon the knockdown treatment, the read coverage at the intergenic region adjacent to the TSS increased, thus indicating possible antisense transcription.*”

9) On page 14 the authors state, "Gene ontology analysis of R-loop peaks identified specific categories of genes regulated by DDX5, XRN2 and PRMT5 such as those involved in ATP binding, DNA damage signaling and histone modifications, respectively". Since the changes in R-loops are not accompanied by changes in transcription as determined by RNA-seq, what would the mechanism of R-loop-mediated gene regulation be in these cases? How do the authors think PRMT5, DDX5, and XRN2 regulate genes in these GO pathways in a way that is dependent on R-loops but independent of mRNA level?

REPLY: The Gene Ontology analysis is indeed confusing without appropriate RNA expression. We have deleted the Gene Ontology analysis (old Figure 6 and old Dataset S3).

MINOR:

1) Throughout the results section, the appearance of some figure panels is not in chronological order. Figure 1C is discussed before Figure 1B and similarly Figure 6D appears in the text before

Figure 6B-C. If the text or figures are reorganized, the text will have a better flow. Figures 1D and E belong with Figure 2A-C where gain/loss of R-loop peaks are demonstrated. The authors should consider revising the organization so that Figure 1 has three panels (A-C) and Figure 2 has five panels (A-E).

REPLY: We have reorganized the panels of Figures 1 and 2 for proper flow with the text.

2) Figure S3E- For the Venn diagrams, keep the order of the different knock-down lines the same as in the upper panel (DDX5, XRN2, and PRMT5) for consistency.

REPLY: We reordered the Venn diagrams for consistency.

3) The methods section is missing the company from which the S9.6 antibody and protein A/G beads are purchased.

REPLY: The methods section was edited as follows: *“DNA/RNA hybrids from 4.4 µg digested nucleic acids, treated or not with RNase H, were immunoprecipitated using 10 µg of S9.6 antibody (ATCC) and 50 ml of protein A/G agarose beads (Sigma) at 4°C for 2 h or overnight in IP buffer (10 mM NaPO₄, 140 mM NaCl, 0.05% Triton X-100). The beads were then washed four times with IP buffer for 10 min at room temperature, and the nucleic acids were eluted with elution buffer (50 mM Tris–HCl pH 8.0, 10 mM EDTA, 0.5% SDS, and 70 µg of protease K) at 55°C for 1 h.”*

4) The EGR1 genome track is shown twice in Figure 5 and Figure S5. These appear to be identical.

REPLY: Indeed they are identical. We reorganized Figure 5A and old Figure S5 to remove the duplication. See new Figure 5A.

5) The rationale for analyzing "intronic reads" is not clear. Are these representing nascent RNA molecules? A short explanation in the text would be useful.

REPLY: The results section was edited as follows to indicate that intronic reads represent nascent RNAs. The text was edited as follows: *“Analysis of the 46839 DRIP-seq peaks (a subset of the 50650 total peaks) identified without a gain or loss in signal relative to siCTL in any of the samples revealed that more than half of the peaks (62.9%) were located in intronic regions, representing nascent transcripts, while 22.6% were located near defined non-coding and coding genes at the promoter-TSS (transcription start site), exons including 5'- and 3'-untranslated regions (5'-UTR, 3'-UTR), and the TTS (transcription termination site) (Figure 4A; Unchanged pie chart).”*

6) The RefSeq tracks in IGV appear to be "collapsed" where overlapping annotations are shown on top of one another. The track can easily be "expanded" in IGV which would allow for better visualization of the TSS and TTS for genes that are overlapping such as the ones in Figure S1B (SSTR5-AS1 and SSTR5). As is, it is hard to distinguish the two ORFs.

REPLY: We have redone the IGVs of Figure 1C and S1B to properly distinguish the ORFs.

7) The word "significant" is used without implying statistical significance throughout the manuscript i.e pg 12. Please replace these with other adjectives that don't imply statistical significance if the statistical test is not shown as in Figure 5A and S5A.

REPLY: We have removed the usage of “significant” for Figure 5A and S5. We have added our

threshold requirements in Figure 5 and S5 legends as follows: “*Read coverage of DRIP-seq and RNA-seq signal centered at the R-loop gain peak associated to the EGR1, ACTG1, RHOB, RBICC1, SOGA1, STIL and UBALD1 genes and the downstream of IER2 gene loci in siCTL and siDDX5 (absolute log2 fold change > 1 and false discovery rate < 0.1, Wald test).*” and “*Read coverage of DRIP-seq and RNA-seq signal centered at the R-loop gain peak associated to the RPLP1, RPP25L, FBXW5 and PMEPA1 gene loci in siCTL and siXRN2 (absolute log2 fold change > 1 and false discovery rate < 0.1, Wald test) as well as to CUL3, NYAP1, CPT1A and NACC2 gene loci in siCTL and siPRMT5 (absolute log2 fold change > 1 and false discovery rate < 0.1, Wald test).*”

8) Page 8: "Less significant overlap of loss peaks". Since the p-values are extremely low, these differences are certainly still very significant and it should not be referred to as "less significant".
REPLY: The reviewer is correct. We removed 'less'.

9) If the cutoffs for fold change and FDR are marked on the volcano plots in Figure 1D and Figure S3C, it would be easier to interpret the results. As is, the cutoffs are only mentioned in the materials section and harder to access.

REPLY: The fold change and FDR was added to new Figure 1C legend “*(absolute log2 fold change > 1 and false discovery rate < 0.1, Wald test)*”. Figure legend of Figure S3C was edited as follows “*Volcano plot of the RNA-seq differential expression for the genes lying nearest to the DRIP-seq consensus peaks, highlighting genes that lie near peaks with a gain (red) or loss (blue) in R-loop signal upon the corresponding knockdown treatment. Dashed lines indicate an FDR of 0.05 and absolute log fold change of 1.*”

10) Figure S1A: The legend states the scatter plots represent RNase H treated samples. Is this a typo?

REPLY: Yes it was a typo. We removed RNase H treated.

Reviewer #3 (Comments to the Authors (Required)):

The article by the Richard lab maps R-loops in the genome of cells siRNA depleted for DDX5, PRMT5 and XRN2. This builds on previous work showing that these factors regulate R-loop levels and is a new effort to identify genomic features associated with R-loops for each factor. The data support the model that DDX5, PRMT5 and XRN2 work together to regulate R-loops at some transcription termination sites (TTS). In addition, a new, possibly independent role for DDX5 in regulating antisense transcription at promoters was proposed based on the changes in DRIP profiles for siDDX5 cells. The manuscript is straightforward and primarily a descriptive analysis of DRIP-seq data. Investigation and validation of the new DDX5 role at promoters is limited to a set of RT-PCR experiments but hopefully will stimulate research in the field. Below are some suggestions to improve the article.

Comments:

- Despite the significant overlap of peaks the differences between DDX5, XRN2 and PRMT5 seem to be an important new finding in this study. PRMT5 knockdown seems to elicit way more new peaks than either of the other two knockdowns (true for both gained and lost peaks). Is there

a technical or biological reason for this? For example, 345 DDX5 peaks overlap with PRMT5, which leaves >2000 PRMT5 peaks that are non-overlapping! The discussion seems to suggest a biological rationale of a splicing defect. Is such a splicing defect evident in the RNA-seq data? Part of this discussion could also include the rationale for focussing on DDX5 as opposed to PRMT5 (which seems to cause a stronger effect on the genome).

REPLY: We edited our discussion to reduce the focus on DDX5 and we now also include antisense data for XRN2 and PRMT5 (New Dataset S3 and Figure S5). The reason why PRMT5 influences more R-loops than DDX5 or XRN2 is that PRMT5 has other RNA helicases as substrates including DHX9 and DDX21 that contain RGG/RG motifs. Also PRMT5 methylation of RNA binding proteins may influence RNA processing, known to affect R-loops. We have edited the text as follows. *“The DRIP-seq results reveal more R-loops affected by PRMT5 deficiency (2632) than by DDX5 (762) or XRN2 (1059) deficiency. As PRMT5 post-translationally modifies proteins, it is not surprising that PRMT5 has an elevated number of cellular R-loops. RNA helicases known to resolve R-loops containing arginine methylation sites (RGG/RG motifs) and potential substrates of PRMT5 include DHX9 (Chakraborty et al., 2018; Cristini et al., 2018) and DDX21 (Song et al., 2017) and there likely others. These observations suggest PRMT5 may regulate other RNA helicases, besides DDX5, to resolve R-loops. Moreover, PRMT5 methylation of RNA polymerase II C-terminal domain leads to the recruitment of RNA helicase Senataxin to resolve R-loops (Zhao et al., 2016). In addition to RNA helicases, PRMT5 methylates many RNA splicing and processing factors (Guccione and Richard, 2019), where defects in these processes also lead to R-loop accumulation (Bhatia et al., 2014; Li and Manley, 2005). The intronic DRIP-seq reads observed for PRMT5 were highly clustered before the TTS and these represent nascent RNAs consistent with the role of PRMT5 in pre-mRNA splicing (Fong et al., 2019; Friesen et al., 2001). Thus it is possible that PRMT5 as well as DDX5 and XRN2 may also influence R-loop accumulation by also affecting the processing and nuclear export of RNAs.”*

- Some of the numbers do not appear to match or are presented in a confusing way. For example, on page 8, 762 DDX5 peaks are identified, but in the comparison to RNA-seq on Page 9 it says "...of the siDDX5 697 R-loop gain peaks...". Which 697? Is this a subset of the 762 or a different set?

REPLY: We corrected the text to clarify that while there are 762 gain peaks, these lie near 697 genes (i.e. some genes have more than one R-loop peak).

- The siPRMT5 induced R-loops did not appear closer to neighboring genes. Is this a statistical artifact because there are so many more PRMT5 peaks?

REPLY: We added *p*-values of statistical significance to Figure 3A and we edited the text as follows *“The 2632 R-loop gain peaks of siPRMT5 were closer to a neighboring gene compared to the unchanged R-loop bulk peaks ($p < 2.22e-16$, Student's *t*-test, Figure 3A). However, the PRMT5 R-loop loss peaks showed similar distances to neighboring genes than unchanged R-loops ($p < 0.16$, Student's *t*-test, Figure 3A)”*

- Confirming by RT-PCR the changes in antisense transcription is an important result. Is there any data to suggest that DDX5 is physically associated to these TSS in control cells? Do the authors have any insight to how DDX5 is recruited? CHIP experiments with DDX5 could help to answer these questions.

REPLY: As suggested, we now confirm by RT-PCR with stranded specific RT primers the antisense RNAs upstream of the TSS of *EGRI*, *ACTG1*, *RHOB*, *RB1CC1*, *SOGA1*, *STIL* and *UBALD1* in siDDX5 cells (new Figure 5C). In addition, we provide new data genome-wide about XRN2 and PRMT5 R-loops also causing changes in antisense transcription (Dataset S3 and Figure S5). Therefore, it is the R-loop presence that influences the antisense transcription and not merely the presence of DDX5 *per se*. The ChIP data of DDX5 would be difficult to interpret, as its role maybe separate from that of its R-loop role. We have added the following text in the discussion about the recruitment of DDX5 near the TSS: “*Thus, DDX5 is likely required near the TSS to resolve R-loops, allowing transcription and minimize DNA damage at these loci. Estrogen-regulated genes were not enriched in our U2OS data (Dataset S1), as these are likely occurring in a tissue specific manner in breast cancer cell lines (Stork et al., 2016) and how DDX5 gets recruited at these sites is not known. Furthermore, whether these R-loops represent proximal promoter pausing sites remains to be determined. Importantly, the overexpression of DDX5 in breast cancer has been observed, validating the therapeutic value of targeting the helicase activity of DDX5 (Mazurek et al., 2012).*”

- Have the authors considered analysis promoter-proximal pause sites? I am not sure if a pausing index dataset is available in U2OS cells but since DDX5 binds to POLII, it may present an opportunity to give more insight into the regulation of spurious antisense transcription in cells lacking DDX5.

REPLY: We did not identify a promoter-proximal pause site dataset in GEO for U2OS cells. We have added the following text in the discussion as follows to address this possibility: “*Estrogen-regulated genes were not enriched in our U2OS data (Dataset S1), as these are likely occurring in a tissue specific manner in breast cancer cell lines (Stork et al., 2016) and how DDX5 gets recruited at these sites is not known. Furthermore, whether these R-loops represent proximal promoter pausing sites remains to be determined.*”

- I was not convinced by the GO analysis in Figure 6. The formation of R-loops is governed by genomic position, GC content, transcription frequency, etc. Its not clear to me that the changes in R-loop position will have functional consequences for the target genes identified or if the enrichments are basically coincidental because of physical features of the genes. There also does not appear to be a consistent change in gene expression (Fig. 6D) based on the R-loop gains. The discussion states that "DDX5 is likely required near the TSS to resolve R-loops, allowing transcripton..." - if true then the shouldn't DDX5 R-loop gained genes be repressed?

REPLY: The Gene Ontology analysis is indeed confusing without appropriate RNA expression. We have deleted the Gene Ontology analysis (old Figure 6 and old Dataset S3).

Minor points:

The original reference for the AQR helicase in the introduction is missing (Sollier et al., Mol Cell. 2014).

REPLY: We have added the reference.

July 22, 2020

RE: Life Science Alliance Manuscript #LSA-2020-00762R

Prof. Stephane Richard
McGill University
Departments of Oncology and Medicine
Lady Davis Institute for Medical Research
3755 Cote-Ste-Catherine Road
Montreal, Quebec H3T 1E2
Canada

Dear Dr. Richard,

Thank you for submitting your revised manuscript entitled "Genome-wide Rloop analysis defines unique roles for DDX5, XRN2 and PRMT5 in DNA/RNA hybrid resolution". We would be happy to publish your paper in Life Science Alliance pending final revisions necessary to meet our formatting guidelines.

- please take a look at our manuscript preparation guidelines and order your manuscript sections accordingly
- please make sure that the author order in the manuscript matches the author order in our system
- please add an ORCID ID for the corresponding author-you should have received instructions on how to do so
- please add Author Contributions and a Conflict of Interest statement to the main manuscript text
- please upload your main and supplementary figures as single files
- please add a callout for Figure S4E
- please upload your tables as editable doc or excel files
- please upload your manuscript as an editable doc file
- please make sure that the font in your figures is big enough to be legible (Figure 1C, Figure 4, Figure 5, Figure S1, Fig. S2, Fig. S5)

A. FINAL FILES:

B. MANUSCRIPT ORGANIZATION AND FORMATTING:

Sincerely,

Reilly Lorenz
Editorial Office Life Science Alliance
Meyerhofstr. 1

69117 Heidelberg, Germany
t +49 6221 8891 414
e contact@life-science-alliance.org
www.life-science-alliance.org

Reviewer #2 (Comments to the Authors (Required)):

The study constitutes a thorough description of DRIP-seq data identifying DDX5-, XRN2- and PRMT5-dependent R-loops genome-wide in human cells. The data provide unique insights into R-loop regulation across the genome and the datasets will be a useful resource to the R-loop community.

Through the conducted revisions, the authors have addressed all comments. No major new changes recommended.

There is only 1 minor point pertaining to the legend of Figure 5B. The legend describes error bars that seem to be missing from the bar graph.

Reviewer #3 (Comments to the Authors (Required)):

The authors have satisfactorily addressed my concerns from the first round of review. I have no additional critiques. The revised manuscript is improved and presents a clearer story.

July 24, 2020

RE: Life Science Alliance Manuscript #LSA-2020-00762RR

Prof. Stephane Richard
McGill University
Departments of Oncology and Medicine
Lady Davis Institute for Medical Research
3755 Cote-Ste-Catherine Road
Montreal, Quebec H3T 1E2
Canada

Dear Dr. Richard,

Thank you for submitting your Resource entitled "Genome-wide Rloop analysis defines unique roles for DDX5, XRN2 and PRMT5 in DNA/RNA hybrid resolution". It is a pleasure to let you know that your manuscript is now accepted for publication in Life Science Alliance. Congratulations on this interesting work.

DISTRIBUTION OF MATERIALS:

Again, congratulations on a very nice paper. I hope you found the review process to be constructive and are pleased with how the manuscript was handled editorially. We look forward to future exciting

submissions from your lab.

Sincerely,

Reilly Lorenz
Editorial Office Life Science Alliance
Meyrhofstr. 1
69117 Heidelberg, Germany
t +49 6221 8891 414
e contact@life-science-alliance.org
www.life-science-alliance.org